# Limited long-term cooling effects of Pangaean flood basalt weathering

Jack Longman [1,2] ✉, Benjamin J. W. Mills [3] & Andrew S. Merdith[3,4]

The emplacement of large igneous provinces (LIPs) is known to be a driver of climate change in Earth's past. However, the balance of climate warming through $CO_2$ emission and cooling through weathering is poorly understood. To better understand the role of LIP emplacement on long-term climate change, here we utilize the SCION coupled climate-biogeochemical model which considers the impact of LIPs through degassing of $CO_2$ and enhancement of local continental weathering rates. This approach results in better correspondence between the modelled output and proxy reconstructions of the period (especially for seawater Sr isotope composition) when compared to previous modelling studies. Of the seven LIPs during the breakup phase of Pangaea (approximately 300–150 Ma), only the Central Atlantic Magmatic Province (CAMP) drives noticeable long-term cooling in the model, a minor effect (between 1-2 °C) despite emplacement of a very large surface area in the humid tropics. Similarly, only the CAMP imparts a significant change in the long-term Sr isotope record whereas the other LIPs of this period do not. Due to limited areal extents, and emplacement outside tropical weathering zones, we conclude most LIPs have no significant global cooling effect on multi-million year timescales.

The emplacement of large igneous provinces (LIPs[1]), systems of voluminous mafic magmatism related to processes other than seafloor spreading, has occurred regularly through Earth's history[2–4]. The release of large quantities of greenhouse gases (up to $1.7 \times 10^5$ $GtCO_2$ in the case of the Siberian Traps[2]), and the emplacement of considerable volcanic terranes associated with LIPs are known to impact many global biogeochemical cycles[3,5,6]. Greenhouse gas release may cause rapid warming[7], with large-scale sulfate release driving short-term cooling[8]. As a result, linkages between LIP emplacement and large-scale changes in the Earth system (e.g., environmental and climatic shifts), are often made[1,3,9], convincingly linking LIP emplacement to a number of mass extinctions[2,3,10–12]. In addition to their potential importance in geologically rapid perturbations of the Earth system such as those observed during mass extinctions, LIPs may be important for setting the climate state of Earth over longer time periods[13], potentially as drivers of long-term cooling[13,14].

The period between 300 and 150 million years ago (Ma), when the supercontinent Pangaea began to rift and break apart, initiated many of the Earth's systematic and evolutionary upheavals that led to the planet's current configuration. During this time some of the largest LIPs in Earth's history were emplaced, sometimes coinciding with mass extinctions. These include the Siberian Traps (252 Ma), which is the largest continental LIP by volume, and widely thought to be the driver of the End-Permian Mass Extinction[7,15–17]. Later in the Mesozoic, the Central Atlantic Magmatic Province (CAMP; 201 Ma), the largest continental LIP by area, was linked to the end-Triassic extinction[18–21]. Further, the Karoo and Ferrar LIPs (183 Ma) have been implicated in the end-Pliensbachian extinction, and Toarcian anoxic event[22–25], respectively.

It is generally assumed that one of the most damaging impacts of LIPs on the biosphere is carbon-rich volatile release (especially CO, $CO_2$, and $CH_4$) enhancing the greenhouse effect and resulting in

[1]Department of Geography and Environmental Sciences, Northumbria University, Newcastle upon Tyne, UK. [2]Marine Isotope Geochemistry, Institute for Chemistry and Biology of the Marine Environment (ICBM), University of Oldenburg, Oldenburg, Germany. [3]School of Earth and Environment, University of Leeds, Leeds, UK. [4]School of Physics, Chemistry and Earth Sciences, University of Adelaide, Adelaide, Australia. ✉ e-mail: jack2.longman@northumbria.ac.uk

catastrophic global warming[3,10,11,26]. This is especially true for examples of LIPs that were emplaced into organic carbon-rich sediment, such as the Siberian Traps[17,27]. However, research has also shown how large volcanic episodes can lead to long-term global cooling[14,28,29]. This cooling can occur via the supply of mantle- or crustal-derived nutrients to the oceans leading to enhanced carbon sequestration via the biological pump[29,30], and through an enhanced silicate weathering cycle fuelled by highly weatherable volcanic rocks[5,28,31]. Consequently, the full impact of LIP emplacement on climate is uncertain, and very few modelling studies consider concurrently the likely cooling and warming impact of LIPs together[13,31]. As such, the exact balance and interplay between LIP-driven cooling and warming mechanisms are unclear[32]. For example, recent work that compiled the area of LIPs within low latitudes over the Phanerozoic concluded that they were not a primary control on greenhouse–icehouse cycles[33], but other recent work argues that the Neoproterozoic Franklin LIP was a critical factor in initiating snowball Earth[34].

To determine if major LIPs could have led to cooling on multimillion-year timescales, and to investigate the cumulative impact of numerous LIP emplacements, we use a long-term climate-biogeochemical model (SCION[31,35]) integrated with the record of LIP emplacement between 300 and 150 Ma[33]. SCION uses a 3D emulated climate, which allows us to move beyond simple consideration of latitude bands to consider intersections of LIPs with local temperature, relief and hydrology, which is essential for estimating weathering—especially considering the prevalence of extensive aridity in Pangaea[36]. Furthermore, SCION produces a range of isotopic tracers, as well as predictions for global temperature and $CO_2$ concentration which can be compared to the geological record to test the validity of its predictions[31,35]. By constraining the model against multiple proxy systems it is possible to make clear quantitative tests of hypotheses. As well as incorporating volumes of degassed $CO_2$ during emplacement, these LIPs are added to the 2D model land surface grid as basaltic terranes and interact with local temperature, relief and hydrology to amplify silicate weathering rates (Fig. 1, see the "Methods" section).

Here, we use an updated version of the SCION model, which considers the high weatherability of LIPs alongside their degassing impact for the period 300–150 Ma. Our approach allows us to simultaneously consider the warming and cooling potential of each individual LIP on the Earth system over long timescales. This is especially important during this period of climatic and evolutionary upheaval.

## Results and discussion
### The role of LIPs in cooling the Mesozoic Earth system
The addition of LIPs to the SCION model impacts the carbon cycle and therefore changes the model reconstruction of climate between 300 and 150 Ma (Figs. 2 and 3). By calculating a total carbon balance between the input of carbon from a LIP minus the removal of carbon via LIP weathering (see the "Methods" section, Fig. 2a), we can disentangle the impact of individual LIPs on carbon cycling. To do this, we run a range of model scenarios (Table S1, see the "Methods" section), these include a baseline (the unaltered SCION code), a model with only LIP weathering considered, and scenarios with only degassing, and with weathering and degassing considered. Unless otherwise stated, our discussion below refers only to the output of the scenario 'Weathering & Degassing High' (Table S1, see the "Methods" section).

Our model results identify that on multimillion year timescales, LIP weathering is the dominant driver of LIP-related carbon cycling perturbations, with the majority of the studied time frame characterized by LIP-induced carbon drawdown, as indicated by carbon balance values below zero (Fig. 2a). This makes sense as LIPs become a permanent feature of the weathering environment, whereas $CO_2$ release is short-lived. The suggestion that LIPs may lead to global cooling has been made previously[13,14,37,38], but it has been challenging to quantify the exact balance between LIP carbon emission and removal[32].

At 300 Ma LIPs are a net $CO_2$ sink in the model (Fig. 2a), reflecting the weathering of previously-established terranes. Following the emplacement of the Siberian Traps (252 Ma) there is a slight enhancement of net LIP-related $CO_2$ removal in the model (Fig. 2c), reflecting the weathering of Siberian Traps material[39] (Fig. 3c). However, the location of this LIP in the high latitudes (Fig. 1, Fig. S1) means the silicate weathering feedback is muted, and weaker than if it were emplaced in the tropics[38]. The signature of this weathering is also noticeable in the seawater $^{87}Sr/^{86}Sr$, which in the weathering-only scenario, declines slightly in the period following the emplacement of the Siberian Traps (Fig. 4b). From 220 Ma onwards, global cooling, driven by $pCO_2$ reduction in the period 220–200 Ma is reconstructed by SCION (Fig. 4d). However, this is also a feature of the 'baseline' model scenario, so is not related to LIP activity.

The CAMP, emplaced from 200 Ma onwards in the model, leads to the highest LIP-related carbon burial values in the Mesozoic (Fig. 2c). The location of the CAMP across the equatorial region (Figs. 1 and 3) and adjacent to an incipient ocean basin, means it is subjected to the most intense chemical weathering regime[35]. The impact of this weathering is most clearly seen in the model seawater $^{87}Sr/^{86}Sr$ record for the 'weathering only' scenario (Fig. 4b), which diverges considerably from the baseline SCION run (scenario 'baseline' in Fig. 4b) and is much closer to proxy reconstructions (Fig. 4b). Intense LIP weathering leads to effective $CO_2$ drawdown through silicate weathering (Fig. 3c) and associated carbonate deposition, and also through nutrient delivery to oceans coupled with a strong biological pump, resulting in higher levels of organic carbon burial in the model. The action of both these carbon sinks leads to slightly cooler climate between 200 and 180 Ma relative to baseline SCION (Fig. 4), but it is noticeable that this shift is only around 1 °C (Figs. 3a and 4c), echoed by a small decrease in $pCO_2$. This shift may go a small way to reconciling the early Jurassic cooling, an event which has proven enigmatic to explain[40,41]. However, cooling of up to 5 °C, as seen in some proxy records in the early Jurassic[40] is not reproduced by our model. It is possible that we underestimate the level of weatherability enhancement represented by basaltic LIP emplacement, meaning we underestimate the cooling impact of LIPs. However, the good correspondence between the seawater $^{87}Sr/^{86}Sr$ record and our model results for the scenario 'weathering and degassing high' suggests the amount of additional unradiogenic material being input to the oceans is of the correct order of magnitude, at least for the period 250–180 Ma (Fig. 4b). Indeed, to reach 5 °C of cooling following the CAMP emplacement, the weathering enhancement value must be raised to 30-fold (Fig. S2b), a value which invalidates the Sr isotope curve, resulting in values well below the reconstructed proxy data (Fig. S3). A weathering enhancement of 15-fold remains possible given the Sr isotope curve across the CAMP (Fig. S3), suggesting it may be a plausible maximum value for this LIP (and one which results in 2 °C of cooling), but this would require considerable uplift of basaltic material. A weathering enhancement value of seven-fold, as used by previous studies[13,31], therefore appears suitable, suggesting LIPs—even the areally extensive CAMP—are unable to cause considerable cooling on multimillion year timescales. From 200 to 150 Ma, the ongoing influence of CAMP basalt weathering is evident in the modelled outputs, keeping global temperatures and $pCO_2$ low despite the emplacement and degassing of two other major LIPs. These LIPs (the Karoo-Ferrar and the NW Australian) have only short-term impacts on temperatures and $pCO_2$ (Fig. 4).

One of the clearest impacts of the addition of LIPs to the SCION model is on the Sr isotope system. Prior to the emplacement of the Siberian Traps, the model is unable to reproduce most of the Mesozoic trends in oceanic Sr isotope composition (Fig. 4). This offset has previously been linked to the poor availability of surface lithological data[35,42], and so the inclusion of LIPs as sources of Sr with igneous signatures more strongly reconciles model outputs with the geological

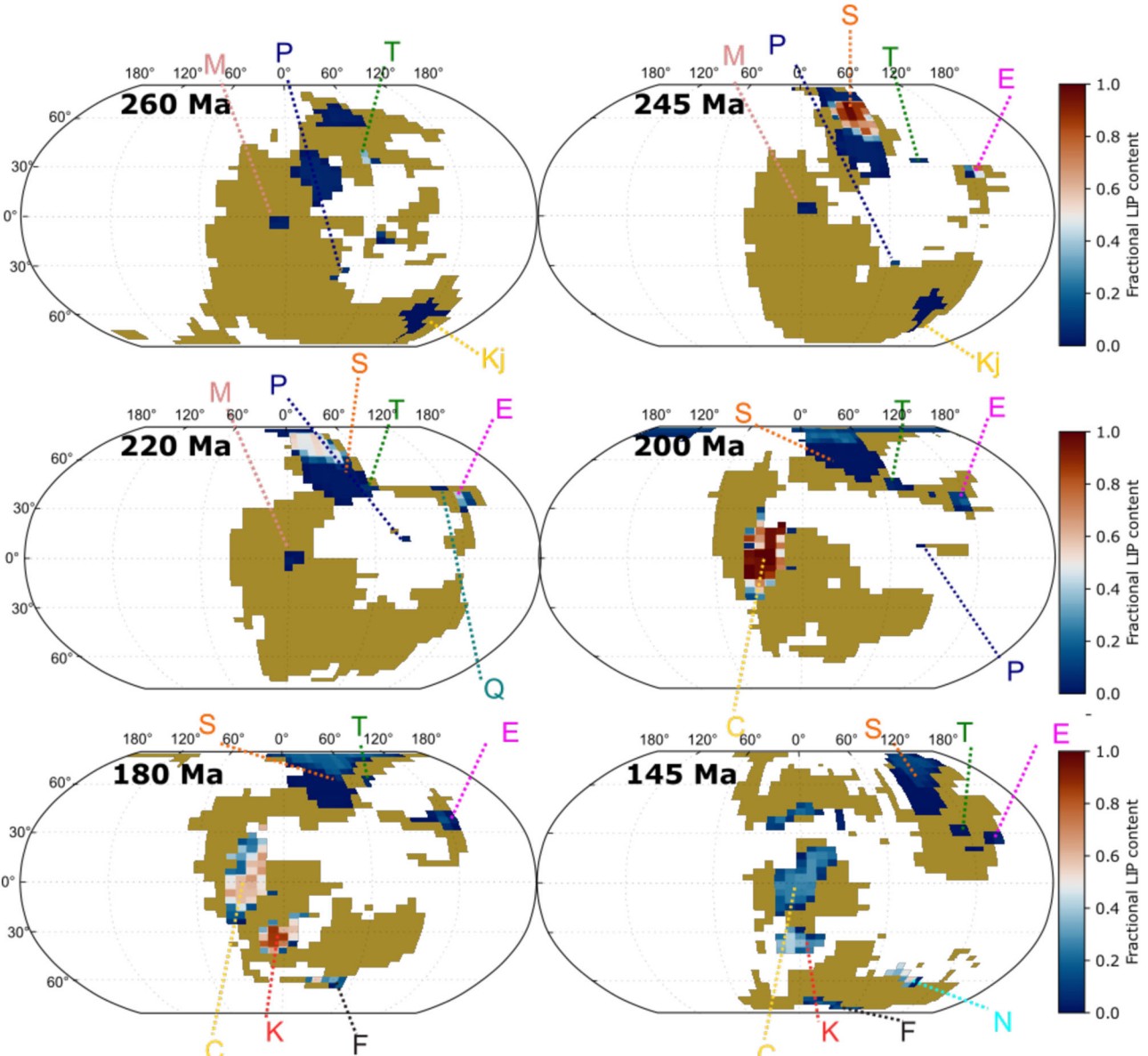

**Fig. 1 | Location of large igneous provinces (LIPs) considered in this study.** Each panel demonstrates the location of each LIP at each time point. Initials pertain to the following LIP names: Magdalen (M), Panjal (P), Tarim (T), Kalkarindji (Kj), Siberian Traps (S), Emieshan (E), Qiangtiang (Q), CAMP (C), Karoo (K), Ferrar (F) and NW Australia (N). Ocean is coloured white whilst mustard is used to highlight the land cover on each map. Overlain are LIP locations which are coloured by their fractional cover, from blue (very low LIP cover) to red (complete LIP cover). These fractional values are used to calculate weathering fluxes for each grid square (see the "Methods" section).

record. In particular, the long-term drop in seawater $^{87}Sr/^{86}Sr$ after 200 Ma, after the CAMP emplacement, is well produced by our updated SCION model. Here, the drop is linked to enhanced basaltic weathering from the emplacement of the LIP at this time (Fig. 4b), where this basalt has a mantle-like Sr isotope composition and therefore acts to reduce seawater $^{87}Sr/^{86}Sr$. Further, Sr input resulting from enhanced continental weathering under high global temperatures also improves the comparability of the data to the model outputs. This is particularly clear across the P–T boundary, where $^{87}Sr/^{86}Sr$ rises markedly. This rise is driven by the warming impact of the Siberian Traps and proportional increase in weathering of radiogenic continental crustal (non-LIP) material–as the LIP was emplaced at higher latitudes away from major weathering zones. In general, the model reconstruction of carbonate $\delta^{13}C$ is within the range of proxy reconstructions (Fig. 4a), although as a result of the single-ocean box in SCION (meaning marginal environments are not considered),

there is little variability outside of the major LIP emplacement periods[35].

Outside of the Siberian Traps and the CAMP, for most of the time between 300–150 Ma, there is little evidence of LIPs driving large scale, multimillion year cooling through enhanced weathering in our model, contrary to suggestions that LIP weathering is a long-term driver of cool climate[43,44]. Our results agree with work that finds no correlation between LIP area and ice sheet size[33], and suggests that any correspondence between emplacement and long-term cooling may not be LIP-weathering related. One potential explanation is that the deposition of volcanic ash, known to drive periods of transient cooling[14], may be more important than basalt emplacement as a driver of $CO_2$ removal[30] on long time scales during LIP episodes. For example, during the late Ordovician, ash supply was sufficient to drive up to 3 °C of cooling for up to a million years in a global biogeochemical model[30]. There was no LIP associated with the late Ordovician, but many LIPs

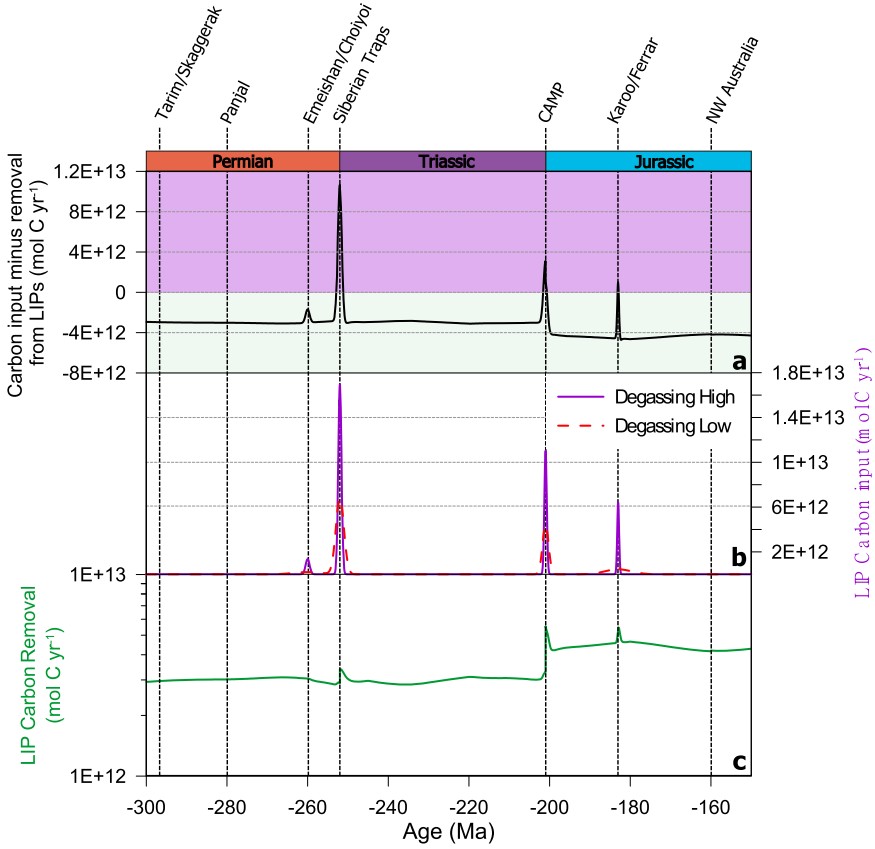

**Fig. 2 | Carbon cycle forcing as a result of Large Igneous Province (LIP) emplacement. a** Overall carbon balance in the 'weathering & degassing high' scenario (Table S1), as a result of LIP input and removal of carbon from the Earth system (see the "Methods" section). **b** Degassing-related input of carbon from LIPs for both 'weathering & degassing high' (purple line) and 'weathering and degassing low' (dashed pink line) scenarios (Table S1). **c** Carbon burial via enhanced silicate weathering from the 'weathering & degassing high' scenario.

had periods of intense explosive volcanism alongside the large-scale effusive eruptions. For example, the North Atlantic Igneous Province in the Cenozoic had an extended episode of basaltic ash deposition[45]. This hypothesis requires further testing, but would explain how LIPs may still be linked with global cooling, despite the apparently small impact of their weathering on global climate.

## LIPs and warming in the Mesozoic

Despite our focus on the cooling impact of LIPs, to consider overall impact we also model contributions to solid-Earth carbon degassing. The SCION model uses the baseline degassing rate of ref. 35 and in this work we add an additional component for contributions from LIPs in three scenarios (see Table S1, see the "Methods" section). Here we reconstruct LIP degassing rates using the timing of LIP emplacement unless the exact period of degassing is known (see the "Methods" section). As such, if pulsed degassing occurred over extremely short timescales such as centuries, as proposed for the CAMP[46], our model would not capture this short-term warming.

As expected, we see transient warming events during the largest LIP emplacements in our study period (Fig. 4c). At the end of the Permian and beginning of the Jurassic periods, the impact of massive, rapid carbon release from LIPs is clear (Fig. 4c, d). Compared to other LIPs in the Mesozoic, the rate and scale of carbon release during the emplacement of the Siberian Traps is nearly an order of magnitude higher (Fig. 2b, ref. 15). In the 'baseline' SCION scenario, transient carbon cycle perturbations from LIPs are not included (Fig. 4, black dashed lines), but in scenario 'weathering and degassing high', the P–Tr boundary is represented by a shift to $p$CO$_2$ as high as 10,000 ppm (Fig. 4d), reflective of proxy data from the period[16]. This is reflected in

the Global Average Temperature (GAT) as modelled in our 'weathering and degassing high' scenario, which rises roughly 7 °C across the P–Tr. However, this rise is well below what has been reconstructed using proxy data, which shows up to 15 °C rise in tropical temperatures (and thus presumably more at the global scale) across the P–Tr[47,48]. This disconnect is potentially related to the low climate sensitivity and relatively high pre-event CO$_2$ within SCION. Later in the Mesozoic, we also see the warming impact of the emplacement of the Central Atlantic Magmatic Province (201 Ma), with a rise of 4 °C in our model (Fig. 4c). This contrasts with reconstructions of above 5 °C change[49] or up to 16 °C change in some proxy data[50]. For the Karoo-Ferrar (182 Ma), we see a rise of 3 °C. Again, this is below proxy reconstructions of between 4 and 7 °C shift across this interval[51,52]. Our model underestimates early Mesozoic (250–200 Ma) temperatures (Fig. 4c), a feature of the original SCION configuration[35], potentially linked to non-carbon aspects of the Earth system such as albedo, or a related to low climate sensitivity of the climate model used (FOAM).

Despite showing significant warming, our temperature results for LIP-induced degassing in the 'weathering and degassing high scenario' tend to be lower than previously suggested[48]. This may be because our degassing rates are controlled by the length of LIP emplacement, and so may result in underestimation of rates if pulsed degassing during the LIP occurred. By running comparative models which reduce the length of time the degassing occurred for to 50 kyr for each event, we can reconstruct changes in climate which are much closer to proxy reconstructions (Fig. S2a). In this scenario ('Weathering & Degassing 50 kyr'), the temperature rise across the P–T is 9 °C, and for the CAMP it is 8 °C, much closer to the reconstructions of 15 °C for the P–T[47,48] and >5 °C for the CAMP emplacement (ref. 49 Fig. S2), suggesting

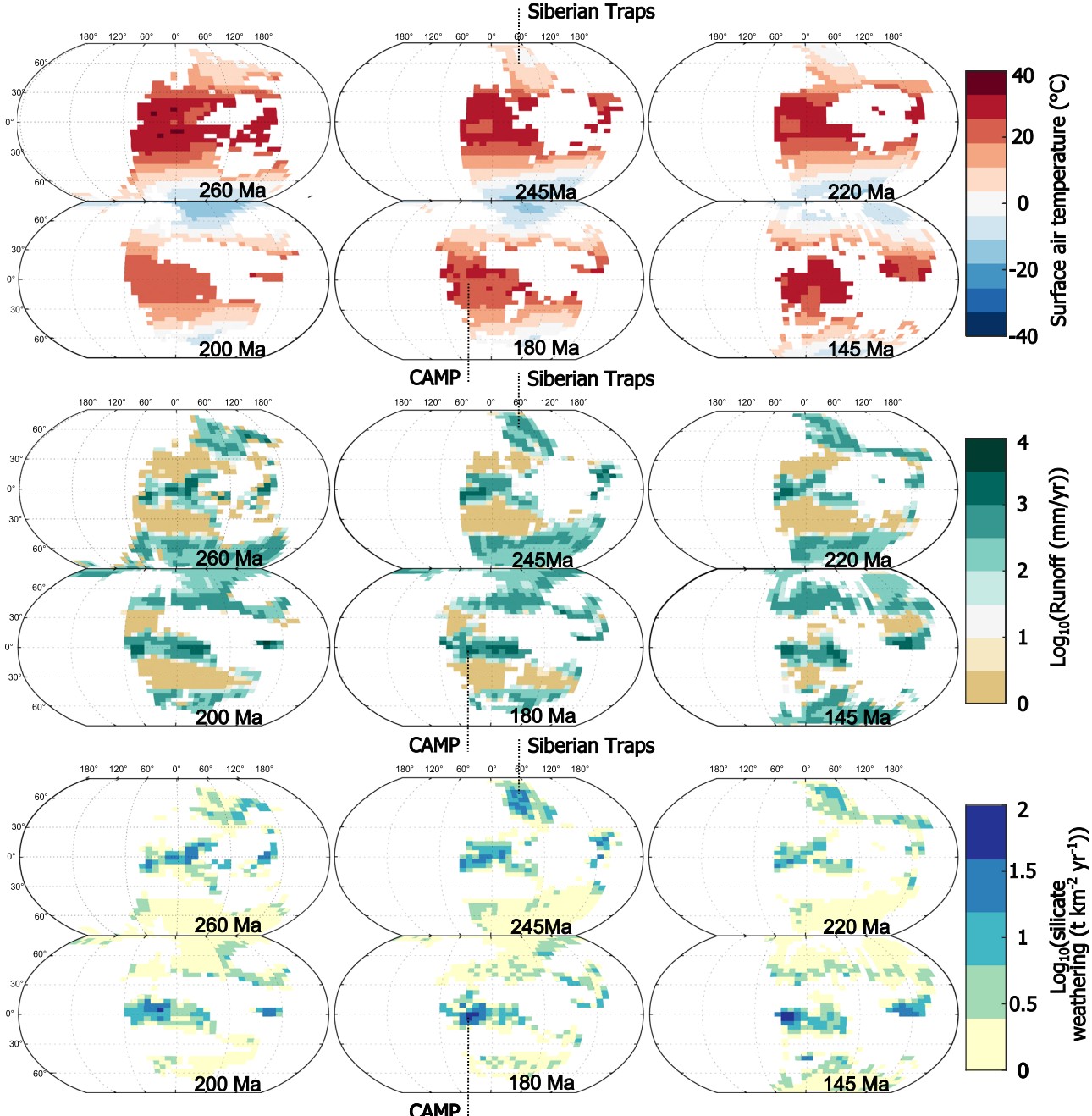

**Fig. 3 | Gridded map outputs for the 'Weathering & Degassing' scenario across the period 300−150 Ma. a** Displays surface air temperature, **b** continental runoff and **c** silicate weathering fluxes. In each map, intensity of the variable from each grid square is indicated via colour bar. The location of the two largest LIPs of the Mesozoic (the Siberian Traps and the Central Atlantic Igneous Province) are specifically highlighted.

carbon volume estimates may be correct, but the periods of degassing may have been much shorter than the lifetime of the LIP. A further model scenario which fully considers the role of cryptic degassing (Weathering & Degassing Cryptic), using the approach of ref. 53, is also able to reconstruct up to 15 °C warming for the P–T (Fig. S2a).

### Implications of limited cooling via LIP weathering

Most studies linking LIPs to global climatic change have focussed on the warming impact of LIPs. However, other studies claim that weathering of highly reactive basaltic terranes associated with LIP emplacement is a driver of global cooling[28,43,44,54]. While we reconstruct the short-term warming in our model, we see little evidence of Mesozoic LIPs driven cooling. The one exception is the emplacement of the

CAMP, which drives global cooling (-1 °C) associated with enhanced organic carbon and carbonate burial and $p$CO$_2$ reduction. We conclude, therefore, that LIPs do not have a major cooling impact on the Earth's climate, even when a spatially-extensive LIP is emplaced in the tropics such as the CAMP (Fig. S1).

A possible link between emplacement of LIPs and glaciation and/or climate cooling has been made in previous studies, but these studies are predominantly correlative, and so testing the feasibility of a causative link over long time periods has not been completed[54]. Our work suggests it is unlikely the Tarim, Panjal, Emeishan or Choiyoi LIPs had important roles in the successive glacial periods of the early Permian (the so-called P1–P4 events[55]), as has been previously proposed[29,56]. We also consider it unlikely that the Karoo-Ferrar LIP drove the late

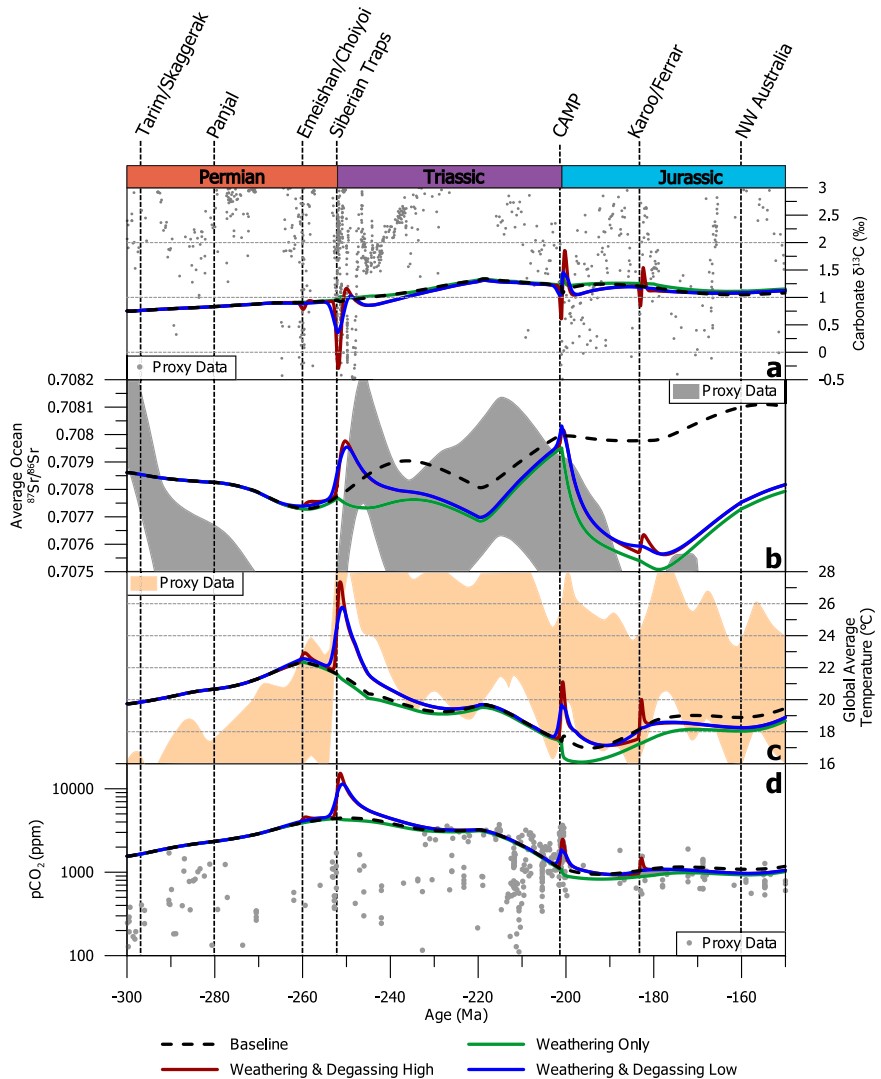

**Fig. 4 | Comparison of model outputs for four different scenarios. a** Model-derived estimates of changing carbon isotope composition, with proxy reconstructions (from ref. 70) in grey circles. **b** Reconstructed oceanic Sr isotope composition compared to oceanic Sr isotope proxy data (grey shaded area[67]). **c** Reconstructed global average temperature compared to reconstruction based on proxy data (orange shaded area[48]). **d** Reconstructed atmospheric $CO_2$ levels (solid lines) compared to proxy measurements of $CO_2$ (grey circles[68,69]).

Pleinsbachian cooling and the Jurassic icehouse episodes[54,57]. Indeed, the emplacement of the largest tropical LIP and the one LIP able to drive global cooling in our model, the CAMP, is not implicated in any global cooling events[33,54,58,59]. However, further tests considering important aspects of the SCION model such as climate sensitivity and the topographic changes related to LIP emplacement must be completed to be certain of these conclusions.

Our findings may also have implications for earlier periods of Earth's history. The emplacement of the Franklin LIP has been suggested as a driver of runaway cooling and eventual Snowball Earth development around 720 Ma[9,60]. However, our results suggest its role could have been minor, even if the Franklin LIP was emplaced as a high-relief terrane in a humid and tropical Neoproterozoic world[60].

### Modelling and plate reconstruction limitations

Our model framework has its limitations. It uses a relatively low-resolution climate emulator based on the FOAM climate model at 4.5° × 7.5° (latitude by longitude), it also does not explicitly include the process of soil formation or more nuanced aspects of LIP emplacement such as variations in crustal thickness, dynamic topography or extension at the location of emplacement. We argue here that these limitations do not greatly affect our conclusions. As noted above, the model reproduces the drop in the $^{87}Sr/^{86}Sr$ record coincident with CAMP emplacement, indicating that it is broadly correctly solving the mass balance of weathered felsic and mafic material at this time. If we were to assume that additional mafic weathering and cooling could result from improved climate model resolution or more accurate treatment of soils or relief, then this would imply a further drop in $^{87}Sr/^{86}Sr$ because mafic weathering produces unradiogenic strontium. This further drop in $^{87}Sr/^{86}Sr$ would then not be in agreement with the geological record, as can be seen in the 'weathering & degassing×30' scenario (Fig. S3).

As demonstrated by this study, the location of the LIP material is key to determining the importance of its weathering in global biogeochemistry. As a result, accurate plate and palaeogeographic reconstructions are vital, as the LIP placement is reliant upon them. In particular, paleolatitude and paleotopography is central to the determination of a LIP's location within or outside of a tropical or mid-latitude rain belt[33]. For example, the muted topographical variation of our plate reconstructions between 220 and 200 Ma may be a factor in the limited weathering of LIP material at this time (Fig. 3c). Therefore, to determine exactly how LIPs drive cooling, work must be completed

to improve plate reconstructions in deeper time and couple them to improved reconstructions of exact extent and location of LIPs. For estimating rates of carbon degassing from LIPs, it is vital for accurate reconstructions of original LIP volume and rate, as well as any cryptic degassing component of LIP emplacement[53]. Considerable work has been completed to understand these features, but due to subsequent erosion and subduction, volume estimates will always be associated with high errors (see Methods and Table S2 for the range of degassing estimates used here). Approaches to reconstruct emplacement rates are improving, and with ongoing methodological improvements in radiometric dating, we anticipate future work to further improve and constrain these estimates.

## Methods

We use the SCION (Spatial Continuous Integration) Earth Evolution model version 1.1 (ref. [35]) in this work but with a number of additions as outlined below. SCION combines a long-term biogeochemical box model with a 3D interpolated steady-state climate and a 2D continental lithology and weathering module. We run the simulation forwards for the whole Phanerozoic but only display the output of the period 300–150 Ma as we intend to focus on the impact of the largest LIPs in Earth history in detail, rather than a general investigation of the entire Phanerozoic. For ease of comprehension, we focus on the outputs of atmospheric $CO_2$, global average temperature, global ocean $\delta^{13}C$ and $^{87}Sr/^{88}Sr$. Model code is available at https://github.com/jlongman1/SCION_LIPs.

### Addition of Large Igneous provinces as highly weatherable terranes

As in previous work[31], we include LIPs as weatherable terranes in the SCION GCM interpolation stacks (see Supplementary Table 1). For this work, we mapped the locations of each LIP onto the palaeogeographic land–sea masks used in the SCION model (Fig. 1). We used the compilation of ref. [33], which is based on two prior compilations[1,3].

In the study of ref. [33], the authors reconstruct the original, full extent of LIPs when and where they were emplaced as a series of digital polygons. They then determined a statistical relationship using the present-day aerial extent of LIPs and age (i.e. time since emplacement) and found an exponential relationship between the data. Reference [33] calculated a LIP half-life of 29 Myr (Eqs. (1) and (2)), allowing us to calculate a fractional area of each LIP at a time after its emplacement ($A_{decay}$):

$$A_{decay} = Ae^{-\lambda \cdot t} \tag{1}$$

$$\lambda = \frac{\log(2)}{T_{1/2}} \tag{2}$$

With $A$ being the original areal content (=1); $\lambda$, the decay constant; $t$, time since emplacement, and; $T_{1/2}$, the calculated half-life (=29 Ma). Note that the half-life increases to 36 Ma if one accounts for the burial of LIPs, but this is not taken into account here. Therefore, at each GCM data-stack timeslice we rasterize the LIP polygons from ref. [33] onto a $40 \times 48$ grid (which is the resolution of the underlying FOAM GCM in SCION) at their correct location. This gives a grid of '0s' and '1s' representing cells of either non-LIP or LIP respectively. We then calculate the fractional content of each LIP cell that has eroded according to Eq. (1) after ref. [33] and update the relevant cell to reflect this (so after emplacement LIP cells will be between 0 and 1, with 1 representing 100% LIP extent). Finally, because the palaeogeographic model used in the SCION model is slightly different from that used by ref. [33], some manual manipulation of the grid cells was required to ensure they were placed correctly with respect to the palaeogeography. We complete this exercise for each individual LIP, so for the

Mesozoic, we have 7 individual LIP maps that vary through time (e.g. Siberian Traps Fig. S4). SCION uses a 'double keyframing' approach to calculate continental processes in between the GCM climate simulation times. For example, at 20 Ma the model is looking at both the 30 Ma and 15 Ma GCM simulations and averaging between them. As most LIPs are emplaced between the keyframe times, we include a 'switch-on' in the code. This checks the time of emplacement for each LIP, and when the model has reached that point, the LIP appears as a weatherable terrane in both the previous and next GCM palaeogeography. See Supplementary Text for further information on the LIP digitization approach.

The model considers all land other than LIPs to be a homogenous mixture[35], which is calculated to be 7 times less weatherable for silicates than LIP basalts. This estimate is based on previous research indicating the weatherability of mafic rocks to be around seven times greater than that of silicates[61] and is the same factor as used in previous work[31,38]. As we are adding large areas of weatherable material to the model, we remove the original global basalt weathering curve outlined in ref. [35]. As such, we assume that LIPs represent most of the total basalt emplaced through the period. The equations governing the relationship between temperatures, runoff and erosion are those used in ref. [35], and we refer the reader to that publication for more detail. The LIPs themselves have no topographic height so the model topography is not altered by LIP emplacement.

### Addition of LIP degassing

As with previous work, we also consider the potential for LIP emplacement to release significant quantities of $CO_2$ and $CH_4$ into the atmosphere[31]. To estimate the amount of carbon emitted from each LIP, we use published estimates of the volume (in $km^3$) of magma emplaced and the duration of the LIP (in kyrs) to produce an estimate of magma emplacement in $km^3/kyr$. We then convert these values to an estimate of carbon degassing using a conversion factor. For silicic LIPs, we use the same approach as[29], which calculates an emission rate of $10^{11}$ g C per $km^3$ of magma emplacement, based upon an estimate of 500 ppm $CO_2$ in the pre-eruptive magma[62]. For basaltic LIPs, we assume pre-eruptive $CO_2$ content of 0.5 wt%, and so an emission rate of $10^{12}$ g C per $km^3$. For modelling purposes, these values are converted to molar carbon flux. For each LIP, we derive two degassing scenarios, 'low' (calculated using the lowest volume and longest duration in the literature) and 'high' (calculated from the highest volume and shortest duration in the literature). Some LIPs are emplaced into carbon-rich rocks, a process that leads to additional carbon degassing through thermogenic alteration. Where estimates of this process are available, for the CAMP[18], Karoo[63] and Siberian Traps[64], we use those values for our 'high' scenario (see Table S1 for further information).

For each LIP, the rate estimate is used to emit $CO_2$ into the model atmosphere. We use Gaussian curves to complete this, with the midpoint of the LIP activity and the peak of carbon emissions used to construct the function (see Table S2). We set the width of the Gaussian function to be related to the period of activity known for the LIP. For the 'low' scenario, we construct the curve based on the longest possible duration, and for 'high' the shortest. For example, the Siberian Traps are taken to have degassed in 2 Myr[65] in the 'low' scenario and in 0.8 Myr[66] in the 'high' scenario, see Supplementary Table 1 for all LIP details. In addition to the 'low' and 'high' scenarios, we run a further configuration with the full impact of cryptic degassing of the Siberian Traps included, using the approach of ref. [53]. This configuration ('high & cryptic') considers a more complex approach to estimating the total degassing of a LIP including degassing that continues after the emplacement of lava has finished[53]. For carbon isotopic mass balance, we include a new 'LIP $CO_2$ $\delta^{13}C$' estimate in the model, which we take to be −5‰, and is the isotopic composition of the degassed $CO_2$. All modelled configurations are detailed in Table S1, and the ranges of input data relating to these configurations are in Table S2.

## Model scenarios

The aim of this work is to investigate the holistic role of LIP emplacement on the Earth system. However, we also test the dependence of the model upon each individual factor associated with LIP emplacement. For this, we construct several model scenarios. The first, which we term 'Baseline' is SCION version 1.1 without any model additions. Second is the model with no enhanced degassing, but with the enhanced weatherability of LIPs included ('Weathering Only'). Additionally, we run three scenarios, with only degassing and no enhanced weatherability (LIP weatherability set to the same as homogeneous silicates). Degassing in these scenarios varies with the first utilizing the 'low' estimates ('Degassing Low Only'), the second using the 'high' estimates ('Degassing High Only'), and the third with the 'high and cryptic' estimates ('Degassing High & Cryptic Only').We also run these three scenarios again, but with LIP weatherability on ('Weathering & Degassing Low', 'Weathering & Degassing High' and 'Weathering & Degassing High & Cryptic'). An additional test runs a model where all degassing episodes are assumed to have only 50 kyr duration (an arbitrarily short time to drive rapid carbon degassing rates), to investigate the ability of rapid carbon release to drive climatic warming (scenario "Weathering & Degassing 50 kyr"). All model scenarios are outlined in Table S1.

For the 'Weathering & Degassing High' scenario we perform sensitivity tests, completed via the running of 1000 simulations with random variation of a number of variables following previous work[31,35]. Specifically, the weathering enhancement from plant evolution is varied between no enhancement and 7-fold enhancement, the carbon isotope fractionation associated with microbial sulfate reduction between 20–40‰ and the fractionation associated with photosynthesis between 20–30‰ (land) and 25–35‰ (marine). See ref. 35 for further details on SCION sensitivity tests. Results of the sensitivity analysis can be found in Fig. S5.

To validate the model results, we compare a range of geological and proxy data. We generally select the data based on two primary factors. Firstly, the record covers the entirety of our modelled period, and secondly that it is global in scale. Using these criteria, we compare our model outputs to the global average temperature reconstruction of ref. 48, the global seawater $^{87}Sr/^{88}Sr$ curve of ref. 67 and compilations of atmospheric $CO_2$ (refs. 68,69) and carbonate $\delta^{13}C$ (ref. 70).

## Carbon balance calculations

We use the mean values of the 'Weathering & Degassing High' scenario to calculate an overall carbon balance. That is the total impact of LIP emplacement on carbon cycling. For this we subtract C associated with basalt weathering (basw in the model) from the LIP degassing rate (LIP_CO2 in the model outputs), resulting in a value of carbon perturbation associated with LIP emplacement at each time step (Fig. 1a).

## Data availability

All data generated or analysed during this study are included in this published article (and its supplementary information files) and can be generated using the code detailed in the 'Code availability' section.

## Code availability

SCION model code is available at https://github.com/bjwmills/SCION. The code version used in this manuscript (ref. 71), which can be used to produce all data presented here, is available at https://github.com/jlongman1/SCION_LIPs.

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

## Acknowledgements
B.J.W.M. and A.S.M. acknowledge funding from UKRI grant NE/X011208/1. A.S.M. acknowledges ARC DECRA Fellowship DE230101642.

## Author contributions
J.L., B.J.W.M. and A.S.M. conceived the research, developed the methodology and the investigation. J.L. made the figures and wrote the first draft of the paper. All authors contributed to and approved its contents.

## Competing interests
The authors declare no competing interests.
