## [Peer Review File · Nature Communications]

Limited long-term cooling effects of Pangaeian flood basalt weathering

Corresponding Author: Dr Jack Longman

Version 1:

Reviewer comments:

Reviewer #1

(Remarks to the Author)

Review Longman et al.: Limited long-term cooling effects of flood basalt emplacements

The concept of the study is interesting and novel, and based on that, I would recommend publication. However, I do think the authors could improve the manuscript, and I have a series of questions and comments that I believe the authors should address. To summarize, the list below represents my main concerns:

1. The original pre-erosive extent/volume of any LIP is highly uncertain, and so is the magnitude of degassed carbon. I can't see that the authors really discuss this, and take these uncertainties into account in their model. As it is very difficult to constrain carbon release from LIPs, a range of potential values for each LIP is often shown by different studies/publications. The same is true of most data shown in your supplementary Table S1 (i.e. the LIP volume, carbon degassing rate and timing, but also original/pre-erosive LIP volume). I can't really see that authors have taken any ranges into account, but have instead picked one value per LIP. There is no discussion as to why they picked those specific values. Do they consider that value to be more realistic? If so, why is it more realistic? How would your choice of value affect your model results? For example, if you consider minimum and maximum degassing values for each LIP, how would that impact your results? I understand that you have to make a choice, but I believe it would improve the paper if you include a discussion as to why you chose what you did, and how this could impact your model. For example, you could do additional minimum and maximum degassing scenarios (based on the range of published estimates) to test the impact. There are many ways to do this. You can think about it. Even if there is a minor difference, such comparisons could strengthen your study because you show that you have taken different possible scenarios into account.
2. The authors don't take thermogenic carbon release into account. Why not? For LIPs emplaced in carbon-rich host rocks, the magnitude of thermogenic degassing could be equivalent or even larger than the magmatic carbon. Surely this could affect your model results?
3. I find your discussion a bit hard to follow. I think this is partially due to your figures, especially Figure 2 and 4, which I believe could be improved (see specific comments below). Further, I believe your discussion would improve if you could be more specific when linking the descriptions of your results with your figures. For example, please refer specifically to the appropriate figure panel when you discuss your results (for example 2a instead of just Fig. 2). It would also help to specify which curve/data point you are referring to (e.g., yellow line, Fig. 2c).
4. Parts of the discussion include comparisons of your model results to proxy data that are not present in your figures. Why is this? Why not include all available proxy data in your figure? Please discuss why you chose the specific proxy data to compare to, while leaving out others. Also, I feel like there is a lack of thorough discussion regarding why your model results don't always match up with the proxy records shown in Figure 4. There is some discussion regarding the Sr curve, but what about pCO₂, temperature and the C isotopes?
5. It would be very helpful to see an overview of all the different model runs; I suggest to create a summary table. Further, please be consistent with model run names throughout the text and in all figures, and make sure the same model run always has the same plot color/style/type. Combined, this will make it much easier for the reader to follow along.

See specific comments below:

13: I know you are focusing on long-term cooling effects, but perhaps you could at least mention (in the introduction) the potential short-term cooling effect of sulfur release from LIPs. See for example Landwehrs et al. (2020) or Black et al. (2018).

19: Please specify what kind of model you used.

19: Could you please quantify this “minor effect” in your abstract? Include specific quantitative results from your study.

25: I suggest this change: “The emplacement of large igneous provinces (LIPs), systems of voluminous mafic magmatism related to processes other than seafloor spreading, has occurred regularly through Earth’s history²⁻⁴”.

27: I would replace the word “enormous”. Also, please quantify the carbon release; e.g., up to xxx Gt; in the range of xx-xxx Gt C.

29: I suggest this change: “As a result, linkages between LIP emplacement and large-scale changes in the Earth system (e.g., environmental and climatic shifts) are often made^{3,7-8}, convincingly linking LIP emplacement to a number of mass extinctions^{2,3,9-11}”.

43: I would use Davies et al. (2017) instead of Schaller et al. (2012) in this case. Marzoli et al. (1999) is a classical paper for linking CAMP and the ETE.

44: I would also cite Greber et al. (2020).

46: I don’t agree with this statement – the consensus is that carbon release is important, but cannot explain the mass extinctions alone. I would thus moderate this statement, and avoid saying that C release is a primary kill mechanism, because we actually don’t know that.

48: LIP emplacement into carbon-rich sediment is not really that rare; some other examples besides Siberian Traps are CAMP, Karoo, HALIP, Emeishan.

53: I would replace the word “holistic”.

69: I would replace “make sense”.

70-71: Please rewrite this sentence.

74-77: Please rewrite this sentence.

80-81: How do Figs 2-3 clearly show that the inclusion of LIPs alters the model reconstruction of climate? What am I supposed to be looking at? Please be more specific.

81-82: The input of carbon minus the removal from weathering - is this actually shown anywhere? Is this supposed to be what we see in Fig. 2b?

81-84: This sentence seems out of place and like doesn’t belong there. It’s very broad. I would move to introduction or delete.

85-87: I don’t think Fig. 2 highlights the fact that weathering is the dominant driver. There is a lot going on in this figure. Am I supposed to be looking at 2b or c here? Please be more specific.

Some other comments on Figure 2:

- I would make the legend font size bigger.
- What do you mean by “modern area”?
- What do you mean by relative LIP area versus present?
- What is the difference between the 0.05 Ma and 5 Ma degassing curves? This is somewhat explained further down in the discussion/methods, but you refer to Fig. 2 already in the beginning of the discussion, so the reader has no idea what these curves are supposed to represent. Either move them to the supplement, or include a description in the figure legend, or refer to specific text/table where you describe what these are.
- What does “Degassing Data” mean?
- I suggest to use different units of the y-axis for panel b and c. These values are difficult to read and they also make the figure look too busy, and it’s hard to see where panel b/c starts/ends. Do you have to present your results in mol C?
- I don’t completely understand what panel b is supposed to show. Is this supposed to be carbon input minus carbon removal? So, everything in the yellow area represents net removal and the opposite for the purple area? Can you perhaps think of another way to show this?
- Due to the range of y-axis in panel c, it looks like the carbon removal from weathering is constant (yellow line). This is an important part of your results. I think you should “zoom in” so we can see the variability.
- I suggest these following changes to Fig. 2 so that your most important results are highlighted: Move panel a to the supplement. Only show the degassing curve that you focus on in the main text and show the other curves in the supplement. Change the y-axis units of panel b and c. For the yellow curve, show a more appropriate range of the y-axis. For panel b, place y-axis ticks on for example the right side, and then for panel c, show the ticks on the opposite side. Or, could you instead combine the carbon balance in panel b and degassing curve in panel c into the same plot? And then show the carbon removal in a separate panel instead.

92-93: Please refer to a figure to back up this statement.

93-94: Where do I see that there is slight enhancement of net LIP-related CO₂ drawdown this in Figure 2?

93-94: I suggest this change: "Following the emplacement of the Siberian Traps (252 Ma), there is slight enhancement of net LIP-related CO₂ drawdown in the model (Fig. 2), reflecting the weathering of Siberian Traps material³⁶. However, the location of this LIP in the high latitudes (Fig. 1) suggests that the silicate weathering feedback is much weaker than if it were emplaced in the tropics³⁵".

97-98: If the reader does not read the methods section first, they don't what you mean by "weathering-only scenario". I think you should inform the reader about your different scenarios at the beginning of the "Result and Discussion" section. I suggest that you make a summary table that describes your different scenarios. Then you can just mention that you have x number of scenarios and refer to the table, and the reader can get more information there.

99: Specify which panel in Fig. 4 that show these results.

99-100: Which figure(s) show this? What do you mean by "original model"?

102-103: I can't really see that CAMP leads to the highest LIP-related carbon burial values in Fig. 2.

103-104: Figure 2 doesn't show the location of CAMP across the equatorial regions.

110-112: What are you comparing to the baseline here? The weathering only scenario (blue line)? Please be specific. Also, please refer to the specific panel in Fig. 4 that you are describing (in this case 4c I think). It's really hard to see a 1°C shift because of the range of the y-axis in Fig. 4c; this is also the case for Fig. 3: I don't see a shift of 1°C in this figure. Please refer to Fig. 4d at the end of this sentence when you are referring to pCO₂.

114-115: "as seen in some proxy records in the early Jurassic³⁷": Here you refer to Dera et al. (2011), but Fig 4c show proxy data from Scotese et al. (2021). Why do you only show proxy records from Scotese in your figure? But then compare to Dera in the text? This question also goes for the rest of the panels of Fig. 4. They all have just one reference for the different proxy data. Why did you choose these proxy data as opposed to other records in other publications? I understand that you must make a decision, but it would be nice if you could discuss your choices.

117-118: Again – which lines are you comparing? Which model results? Please be specific. Also, I am not sure if I agree with the general statement that "there is good correspondence between the seawater 87Sr/86Sr record and our model results". I see some correspondence for the green and blue lines between ca. 250 and 180, but that period represents only parts of your full study period.

118-119: I don't understand how Fig. 3 shows that "the amount of additional unradiogenic material being input into the oceans is of the correct order of magnitude". Where in Fig. 3 am I supposed to see this?

128-129: You state that the model is unable to reproduce most of the Mesozoic trends in oceanic Sr isotope composition prior to LIP addition and refer to Fig. 4. But don't you add LIP (Tarim/Skagerak) already at 300 Ma, so in the beginning of your study period? So, what do you mean by prior to LIP addition then? I would argue that, generally, your model is unable to reproduce Sr record prior to ca. 250 Ma and after 170 Ma.

130: delete "a".

136-140: Please rewrite this sentence.

146-148: Based on your results, how can you conclude/propose that volcanic ash is more important than basalt emplacement as a driver of CO₂ removal? What is the evidence behind this statement?

151-152: which model run uses this baseline degassing rate? If you make the summary table I suggested above, you could include that information there.

157-158: Please be specific when referring to the figures here; I assume you are referring to Fig. 4c, but which panel in Fig. 2?

158-159: Again, is this Fig. 4c?

159-161: How do I see that the scale of C release during ST emplacement is larger in Fig. 3?

160: Please replace the word "speed".

161-163: There is no red line in Fig. 3. Also, the baseline run should be a black dashed line right (not red)? What do you mean by the "new model scenario"? Please give all model scenarios clear names (list these in the table I suggested), and consistently use the scenario names throughout the text so that it is always clear which scenario/model run you are talking

about.

163-164: I think you are referring to Fig. 4d here? Also, here you refer to proxy data from Wu et al. (2021), but panel d in Fig. 4 includes proxy data from Saltzman & Thomas (2012). Again, why do you compare to proxy data that is not in the figure, and why is this proxy data not in the figure? Why did you choose to present only Saltzman & Thomas (2012) in your figure?

164-165: Are you referring to your own model results here or to another study? Please specify.

170: Again, it is very hard to see a 4°C rise in Fig 4c due to the range of the y-axis.

174-175: Which results? Please be specific. Also, you refer to “previous proposals”, but mention only one reference.

177-179: Please add a more thorough description of this additional run in the methods section. Again, if you make a summary table, you can include information there. You say “reduce the degassing window to 50 ka for each even”; what is the original “degassing window”? And where is this described? I can barely see the dashed cyan line in Fig. 4c (again, please refer to the correct panel). I would suggest to change the color, and/or “zoom in” and change the range of the y-axis. Also, please be consistent, use either 50 ka or 0.05 Ma (as shown in Fig. 2 or in the methods; line 302).

Some other comments on Figure 4:

- Add grey dots and shading to the legend and name it “proxy data”.
- I don't think panel a is discussed at all in the main text? So, you could move this to the supplement (or discuss it, if you think these results are important).
- It's really hard to see the different model results, and compare the different runs, because the range of the proxy data is so much larger than your results (y-axis). I understand that you want to show the range of the proxies, but your model results are the most important. I suggest that you “zoom in” and focus on a y-axis range that is more appropriate for showing your results. This would cut off the proxy data, but you could show the full range in the supplement.
- I recommend to change the cyan color because it is very hard to see, although this problem may be solved if you change the range of the y-axis.
- From looking at this figure, the difference between the two “Weathering & Degassing” scenarios is not clear. If you make a summary table, you could refer to that in the figure caption.
- In the figure caption you say upper panel, below, third and final panel, but why not refer to a,b,c,d?
- Please thoroughly describe in the discussion: 1) why you chose the specific proxy data shown in this figure, and 2) why your model curves don't match the proxy data in all four panels.
- Are the green and cyan lines here equivalent to the red dashed and red dotted lines in Figure 2? Same question for the blue (weathering only) line in Fig. 4 and the yellow line (weathering) in Figure 2. If so, please be consistent with both the color/type of line and the name of the model run.

180: What do you mean by “previous reconstructions”?

183: I would replace the word “purport”.

190: Another important limitation is the uncertainty from original LIP volume/extent and magnitude of carbon degassing from the LIPs, but this is not discussed. Please add that to this part of the discussion.

204-222: These sections seem out of place. You kind of jump back and forth between limitations and implications (the first two and last paragraphs discuss limitations). Perhaps re-structure this section into “implications” and “limitations” separately.

231: Please add a reference.

238: What do you mean by “whole ocean”?

248: I suggest this change: “Ref.30 calculated a LIP half-life of 29 Ma (Equations 1 and 2), allowing us to calculate a fractional area of each LIP at a time after its emplacement (A_{decay}):”. Then after “... the calculated half-life (=29 Ma).” add “Note that the half-life increases to 36 Ma if one accounts for burial of LIPs30, but this is not taken into account here”.

260-262: Just to clarify: does this mean that 1 = the maximum extent of the LIP, and 0 = 100% of the LIP has eroded?

284-287: Instead of using Siberian Traps as an example, list all the LIPs where this is true and you used a value from previous literature. And again, explain why you chose that specific degassing value (as opposed to other estimates, or for example the mean of a range).

287: Also list the LIPs where you calculated this yourself. I know that this information is provided in Supplementary Table 1, but you can easily just list the different LIPs here, and then refer to the table for more information.

298-299: Does all literature really agree that the Siberian Traps and Karoo degassed in 1 and 0.47 Myrs, respectively? There are more recent papers than those cited, for example Burgess et al., (2017), Kasbohm et al. (2021), Greber et al. (2020), Gaynor et al. (2022). The same question goes for the other LIPs. The rate of degassing is important when constraining climatic and environmental effects. Please explain why you chose these specific references/values.

300-302: What do you base the duration of 0.05 Ma on? Why do you refer to Fig. 1 here? Give this model run a specific name, and be consistent throughout the manuscript and in the figures.

315-318: please describe this sensitivity test more thoroughly. What does “random variation of a number of variables following previous work” mean? Which variables? Are the results from this sensitivity test only shown in Sup. Fig 2? What do the results tell us? Sup.Fig.2 is lacking a legend, so it’s hard to understand what I’m looking at here.

332: Mean of what? The 1000 simulations? Does the green line shown in Figure 4 represent a mean of the 1000 simulations?

Figure 3: This figure is barely mentioned. Maybe move to supplement? Also, add a,b,c to the different panels.

Fig. S1: Add legend. I would change the yellow color – it’s quite difficult to see.

Supplementary Text: “This value is then multiplied by the LIP multiplies (in this case 7)”. Note it should be “is” and not “in”. Also, could you please remind the reader why 7?

(Remarks on code availability)

Reviewer #2

(Remarks to the Author)

The impact of Large Igneous Provinces (LIPs) on climate is widely acknowledged in scientific research, yet it remains largely untested. With the potential to cause both warming and cooling effects, LIPs serve as an ideal mechanism to explain puzzling climate shifts. At least two reasons make this study valuable and interesting:

1. It specifically tests the effect of LIPs on the long-term carbon cycle and climate change during a period with multiple LIP episodes.
1. It demonstrates a limited but significant effect, with results validated by strontium (Sr) simulations that align with geological records, providing constraints on its reliability.

This manuscript tackles an essential and engaging topic, numerically assessing the impact of LIPs on both the degassing and weathering components of the carbon cycle. The study’s scientific importance is significant, especially given the widely hypothesized but still underexplored role of LIPs in climate shifts and mass extinctions. Furthermore, it addresses the current knowledge gaps regarding the true climatic potential of LIP events.

In terms of scientific quality, the study is rigorous and well-executed. The authors thoroughly test the reliability of their methods and provide detailed discussions of their approach. While additional tests, such as reverse modeling and climate sensitivity analyses, could further solidify the conclusions, the current methodology is robust and well-supported. The scientific reproducibility of the study is strong, with the methods clearly explained and the data presented in a way that allows for replication.

The data and methodology used in this study are solid, with justifications for the time interval and LIP selection, models employed, and acknowledged limitations. This thorough approach enhances the credibility of the results, making the study a reliable source for further exploration of LIPs’ climate impact.

The analytical approach is equally strong, providing a clear framework for understanding the relationship between LIPs and the long-term carbon cycle. However, some clarity is needed in the methods section, particularly regarding the link between the General Circulation Model (GCM) simulations and the LIP effects, which remains somewhat unclear. Additionally, the effect of the Siberian Traps could benefit from a more nuanced treatment, especially in light of evidence suggesting that its influence may not strictly align with its location outside the tropical zone, indicating a potentially more complex, non-latitudinal distribution of its climatic effects.

The validity of this study is well-established, with its methods, findings, and conclusions all supported by thorough analysis. The presentation is effective, with figures that enhance the text and aid comprehension.

Addressing the minor points mentioned—particularly in improving the clarity and comprehensiveness of the manuscript—will strengthen the overall contribution. In conclusion, I recommend this manuscript for publication with only minor revisions.

Specific comments: (the numbering applies to the version given to the reviewer file)

(abstract) It would be beneficial to highlight that the implementation of LIPs in the model results in a better alignment between simulated Sr levels and the geological record, suggesting that this approach is plausible and reliable.

(65-67) You mention the prevalence of low-latitude aridity in Pangea, yet none of the maps in Figure 3 show significant arid regions developing in the tropics. Is this discrepancy intentional?

(72-74) It remains unclear to me, even after reviewing the entire study, whether changes in topography related to the implementation of the LIPs were incorporated. If so, what is the magnitude of these changes? Additionally, were any GCM simulations rerun for the time slices where LIPs were included? As we know, all the volatiles associated with LIPs could also influence the hydrological cycle.

(81-84) Regarding Fig. 2, how do you define tropical land? Are specific latitudinal boundaries used? It's important to note that during the Pangea period, latitude-based estimates of tropical land may not be accurate due to the continental configuration of central Pangea. Could you elaborate on how this was addressed in your analysis?

(93-96) This assertion holds true only if we assume that climatic belts were similar to those of the present, which may not always have been the case. For instance, examining between 270 & 250 Ma the distribution of bauxites and laterites, which are indicators of 'tropical conditions,' reveals a shift towards higher latitudes, as well as a tendency for these deposits to be located on isolated terranes. On the other hand, evaporites tend to cover most of the tropical region. Could you discuss how this perspective was considered in your analysis? (similar to previous comment I guess).

(115-117) This presents quite a challenge, as you are attempting to define this effect. Could you try to constrain this underestimation? Specifically, how much would you need to increase the weatherability to achieve the 5°C cooling? Some reverse modeling could help clarify this.

(117-119 + 129-131) This is great and needs to be also said in the abstract.

(140) Higher altitudes ? I guess you meant latitudes?

(140-148) While I agree with your conclusions, I think there might be more to elaborate on in this paragraph. I noticed that the 220 Ma and 200 Ma maps included in SCION show little to no topography compared to the other maps. Could this lack of topographical variation pose a problem for the simulations and potentially impact the effects of LIP weathering? You also suggest volcanic ash as a cooling driver. Is it possible to test this in the model as well?

(167-168) Why not conduct additional simulations with varying climate sensitivity to determine if this is indeed the cause of the disconnect?

(174-180) Similar to the approach with climate sensitivity, it would be valuable to assess whether the degassing required to induce such a temperature rise is plausible. This is somewhat addressed by reducing the degassing window, as it effectively increases the rate of degassing per time unit.

(197-199) While you emphasize this point here, it's unfortunate that nothing regarding this aspect is mentioned in the abstract, as it just makes the study's reliability stronger.

(207-212) However, the tests conducted here do not account for changes in climate sensitivity, and some of the maps exhibit very little or no topography. This point may need to be addressed with more nuance.

(218-219) Very true ! it is a very important statement to make.

(223-227) Yet, it is very unlikely this belt remained located $\pm 10^\circ$ NS latitudes (see 270-250 Ma evaporites distribution).

(262-265 +272) Yes, manipulation is required; however, how are temperature and runoff modeled in relation to the LIPs? This aspect is not very clear in the methods section.

(Figure 1) This is a very nice figure, but it feels a bit crowded with names. Perhaps consider using just the first letter in the figure itself and providing the full names in the figure description.

(Figure 3) I have questions primarily regarding the 260 Ma maps. When surface air temperatures reach 30+ degrees Celsius across nearly $\pm 30^\circ$ N/S, you observe very high runoff and weathering. How is this possible?

(Figure S1) What do the yellow and orange colors represent? I assume they indicate the mean and the upper/lower limits from the sensitivity test?

(Figure S2) This is a fantastic figure! It significantly helps in understanding the weathering implementation.

I sincerely appreciated the opportunity to read this study, which I believe makes a significant contribution. Thank you for allowing me to review it.

(Remarks on code availability)

Version 2:

Reviewer comments:

Reviewer #1

(Remarks to the Author)

Second review; Longman et al; Limited long-term cooling effects of Pangaeian flood basalt weathering.

The introduction gives a clear overview of background information, what the authors have done and why. The discussion and description of results is now much easier to follow as the authors have improved both the text and figures. The additional model runs, analyses, table and discussion make this study much more robust and convincing. I would recommend publication. See some minor comments below (line numbers are from the document including the tracked changes):

18: Could you please specify “better” correspondence compared to what?

80-81: I suggest moving the following sentence to methods: “although the LIPs themselves have no topographic height so the model topography is not altered by LIP emplacement”

123: Please check if the reference to Fig. 3b and c is correct for this sentence.

127: Specify the panel letter of Figs. 3,4

136: The addition of Fig. S2 is great! But I recommend some changes:

- Adjust the y-axis of panel b of figure S2 (“zoom in”) so that we can more easily see the temperature changes from the baseline. I get it that you want to show the full proxy range, but we can see that in panel a.
- Combine panel a of S2 and Fig S4 into one figure. That way we can easily compare all model runs. Get rid of all the white space; we want the focus to be on your model results.
- Place the black dashed line on top so we can see the full extent of the baseline run (it’s kind of hidden behind the solid lines as of now). This applies for all these figures + Figure 4 in the main manuscript.
- Combine the a and b panels of figure S3.

150: This is from my previous review and your response:

128-129: You state that the model is unable to reproduce most of the Mesozoic trends in oceanic Sr isotope composition prior to LIP addition and refer to Fig. 4. But don’t you add LIP (Tarim/Skagerak) already at 300 Ma, so in the beginning of your study period? So, what do you mean by prior to LIP addition then? I would argue that, generally, your model is unable to reproduce Sr record prior to ca. 250 Ma and after 170 Ma.

We have adjusted this sentence here, to reflect our findings better. Instead of saying ‘Prior to LIP addition’, we now state ‘Prior to the emplacement of the Siberian Traps...’.

I agree with your response, but I don’t see this change in the revised manuscript (line 150).

156: Please check if the reference to Fig. 4c is correct for this sentence.

196: Please check if the reference to Fig. 3 is correct for this sentence.

220: It’s impossible to compare the ‘weathering’assing high” with the “50 kyr” run because the red line is hidden. Could you please make one of the lines dashed. Also, (I already mentioned this above), but please adjust the y-axis and/or get rid of the white space so it’s easier to see the change in temperature.

231-232: This sentence seems strange (i.e., whilewhen and cleardriven).

247-248: This reads a bit strange, so consider re-writing: “...further tests considering in more detail important aspects...”

271: I don’t really understand what you mean by “falsified by the geological record”, so consider re-writing this.

284: Delete “this” after “features”

Fig 3: Could you mark CAMP and the Siberian Traps with a symbol instead of the dotted line? You could make the symbol transparent so we can still see the colorbar. I wonder if you should also add letters a,b,c (and update the text accordingly).

Line 410: This is from my previous review and your response:

Fig. S1: Add legend. I would change the yellow color – it's quite difficult to see. Legend added here. Colours of lines have also been adjusted.

I don't see these change in the revised manuscript (Figure S1 is now S6).

(Remarks on code availability)

RESPONSE TO REVIEWERS

Please find our responses to the comments below, highlighted in blue. We attach a version of the manuscript with changes tracked. For our responses below, however, we use line numbers pertaining to the updated (changes accepted) manuscript.

REVIEWER COMMENTS

Reviewer #1 (Remarks to the Author):

Review Longman et al.: Limited long-term cooling effects of flood basalt emplacements

The concept of the study is interesting and novel, and based on that, I would recommend publication. However, I do think the authors could improve the manuscript, and I have a series of questions and comments that I believe the authors should address. To summarize, the list below represents my main concerns:

We thank reviewer 1 for their detailed and constructive comments, and have attempted to address all their concerns in the updated version of the manuscript.

1. The original pre-erosive extent/volume of any LIP is highly uncertain, and so is the magnitude of degassed carbon. I can't see that the authors really discuss this, and take these uncertainties into account in their model. As it is very difficult to constrain carbon release from LIPs, a range of potential values for each LIP is often shown by different studies/publications. The same is true of most data shown in your supplementary Table S1 (i.e. the LIP volume, carbon degassing rate and timing, but also original/pre-erosive LIP volume). I can't really see that authors have taken any ranges into account, but have instead picked one value per LIP. There is no discussion as to why they picked those specific values. Do they consider that value to be more realistic? If so, why is it more realistic? How would your choice of value affect your model results? For example, if you consider minimum and maximum degassing values for each LIP, how would that impact your results? I understand that you have to make a choice, but I believe it would improve the paper if you include a discussion as to why you chose what you did, and how this could impact your model. For example, you could do additional minimum and maximum degassing scenarios (based on the range of published estimates) to test the impact. There are many ways to do this. You can think about it. Even if there is a minor difference, such comparisons could strengthen your study because you show that you have taken different possible scenarios into account.

Whilst the focus of this work was on the development of a model which considers the impact of LIP basalt erosion on the Earth system, as we include degassing in the approach we appreciate the importance of estimating these values correctly.

In line with the reviewer's comment, we have adjusted the degassing aspect of the model in the following ways:

- 1. Rather than a single degassing gaussian for each LIP, we now model three scenarios. One for 'low' degassing, one for 'high' degassing' and one for 'high' degassing with a further 'cryptic' component. For clarification of these model configurations (and in response to a following comment), we detail their setup in a new Table S2.*
- 2. To develop these scenarios, we completed a new literature review to determine the highest and lowest published estimates for LIP volume and duration. From these*

values we then calculate the lowest possible LIP degassing rate (scenario 'low') and highest possible (scenario 'high') and show these numbers in an updated Table S1.

3. For the 'high' scenarios, wherever estimates of total degassing rates including thermogenic or cryptic degassing are available, we use these values (see footnotes b-e in Table S1).
4. For the Siberian Traps, we also consider the estimates of cryptic degassing as published in a recent study using the SCION model (Black et al., 2024). This approach is used in combination with 'high' degassing estimates for the other LIPs, in the scenario 'high & cryptic' (Table S2. Figure S2).

The inclusion of three estimates for LIP degassing allow us to include envelopes for Earth system response, and so we have updated all relevant figures and text. None of the data alter the findings substantially, but in the 'high & cryptic' scenario, the warming post-Siberian Traps is considerably longer. The main conclusion of limited cooling post-LIP emplacement is not changed.

We detail our approach in an updated Methods section (L327-355):

“To estimate the amount of carbon emitted from each LIP, we use published estimates of the volume (in km³) of magma emplaced and the duration of the LIP (in kyrs) to produce an estimate of magma emplacement in km³/kyr. We then convert these values to an estimate of carbon degassing using a conversion factor. For silicic LIPs, we use the same approach as²⁶, which calculates an emission rate of 10¹¹ g C per km³ of magma emplacement, based upon an estimate of 500 ppm CO₂ in the pre-eruptive magma⁵⁸. For basaltic LIPs, we assume pre-eruptive CO₂ content of 0.5 wt%, and so an emission rate of 10¹² g C per km³. For modelling purposes, these values are converted to molar carbon flux. For each LIP, we derive two degassing scenarios, 'low' (calculated using the lowest volume and longest duration in the literature) and 'high' (calculated from the highest volume and shortest duration in the literature). Some LIPs are emplaced into carbon-rich rocks, a process which leads to additional carbon degassing through thermogenic alteration. Where estimates of this process are available, for the CAMP⁵⁹, Karoo⁶⁰ and Siberian Traps⁶¹, we use those values for our 'high' scenario (see Table S1 for further information).

For each LIP, the rate estimate is used to emit CO₂ into the model atmosphere. We use Gaussian curves to complete this, with the midpoint of the LIP activity and the peak of carbon emissions used to construct the function (see Supplementary Table 1). We set the width of the Gaussian function to be related to the period of activity known for the LIP. For the 'low' scenario, we construct the curve based on the longest possible duration, and for 'high' the shortest. For example, the Siberian Traps is taken to have degassed in 2 Myr⁶² in the 'low' scenario and in 0.8 Myrs⁶³ in the 'high' scenario, see Supplementary Table 1 for all LIP details. In addition to the 'low' and 'high' scenarios, we run a further configuration with the full impact of cryptic degassing of the Siberian Traps included, using the approach of ref.⁶⁴. This configuration ('high & cryptic') considers a more complex approach to estimating the total degassing of a LIP including degassing which continues after the emplacement of lava has finished⁶⁴. For carbon isotopic mass balance, we include a new 'LIP CO₂ δ¹³C' estimate in the model, which we take to be -5 ‰, and is the isotopic composition of the degassed CO₂. All modelled configurations are detailed in Table S2.”

2. The authors don't take thermogenic carbon release into account. Why not? For LIPs emplaced in carbon-rich host rocks, the magnitude of thermogenic degassing could be equivalent or even larger than the magmatic carbon. Surely this could affect your model results?

We did consider thermogenic degassing in our original work but appreciate our methodology was unclear. In the updated manuscript we have attempted to clarify this and refer the reviewer back to our response to the previous comment.

3. I find your discussion a bit hard to follow. I think this is partially due to your figures, especially Figure 2 and 4, which I believe could be improved (see specific comments below). Further, I believe your discussion would improve if you could be more specific when linking the descriptions of your results with your figures. For example, please refer specifically to the appropriate figure panel when you discuss your results (for example 2a instead of just Fig. 2). It would also help to specify which curve/data point you are referring to (e.g., yellow line, Fig. 2c).

These suggestions and the detailed ones below are very helpful, and we have revised the figures accordingly. Wherever suitable, we also now refer to panels when mentioning figures in-text.

4. Parts of the discussion include comparisons of your model results to proxy data that are not present in your figures. Why is this? Why not include all available proxy data in your figure? Please discuss why you chose the specific proxy data to compare to, while leaving out others. Also, I feel like there is a lack of thorough discussion regarding why your model results don't always match up with the proxy records shown in Figure 4. There is some discussion regarding the Sr curve, but what about pCO₂, temperature and the C isotopes?

As discussed below, we select the proxy data based on a pair of criteria – firstly that they are global in their representation, and secondly that they cover the entirety of the studied period. We only present single proxy records to ensure the presentation of data is as uncluttered as possible. This means many of the studies we cite would not be suitable for comparison, and so we only discuss them in text.

We feel the results we present have all been discussed in text – we focus in depth on temperature and CO₂ change, for example across major LIP emplacement periods. We do now however include further discussion of why temperatures may be underestimated outside of LIP periods (L198-201):

“Our model underestimates early Mesozoic (250 – 200 Ma) temperatures (Fig. 4c), a feature of the original SCION configuration³⁶, potentially linked to non-carbon aspects of the Earth system such as albedo, or a related to low climate sensitivity of the climate model used (FOAM).”

And the d¹³C data in lines 154-157:

“In general, the model reconstruction of carbonate δ¹³C is within the range of proxy reconstructions (Fig. 4a), although as a result of the single-ocean box in SCION (meaning marginal environments are not considered), there is little variability outside of the major LIP emplacement periods³⁶.”

5. It would be very helpful to see an overview of all the different model runs; I suggest to create a summary table. Further, please be consistent with model run names throughout the text and in all figures, and make sure the same model run always has the same plot color/style/type. Combined, this will make it much easier for the reader to follow along.

We have now included a summary table in the supplementary information detailing the configurations of the runs. Please see Table S2. We have also updated the manuscript section ‘Model Scenarios’ (L357-380):

“The aim of this work is to investigate the holistic role of LIP emplacement on the Earth system. However, we also test the dependence of the model upon each individual factor associated with LIP emplacement. For this, we construct several model scenarios. The first, which we term ‘Baseline’ is SCION version 1.1 without any model additions. Second is the model with no enhanced degassing, but with the enhanced weatherability of LIPs included (‘Weathering Only’). Additionally, we run three scenarios, with only degassing and no enhanced weatherability (LIP weatherability set to the same as homogeneous silicates). Degassing in these scenarios varies with the first utilising the ‘low’ estimates (‘Degassing Low Only’), the second using the ‘high’ estimates (‘Degassing High Only’), and the third with the ‘high and cryptic’ estimates (‘Degassing High & Cryptic Only’). We also run these three scenarios again, but with LIP weatherability on (‘Weathering & Degassing Low’, ‘Weathering & Degassing High’ and ‘Weathering & Degassing High & Cryptic’). An additional test runs a model where all degassing episodes are assumed to have only 50 kyr duration (an arbitrarily short time to drive rapid carbon degassing rates), to investigate the ability of rapid carbon release to drive climatic warming (scenario ‘Weathering & Degassing 50kyr’). All model scenarios are outlined in Table S1.

For the ‘Weathering & Degassing High’ scenario we perform sensitivity tests, completed via the running of 1000 simulations with random variation of a number of variables following previous work^{32,36}. Specifically, the weathering enhancement from plant evolution is varied between no enhancement and 7-fold enhancement, the carbon isotope fractionation associated with microbial sulfate reduction between 20–40‰ and the fractionation associated with photosynthesis between 20–30‰ (land) and 25–35‰ (marine). See ref.³⁶ for further detail on SCION sensitivity tests. Results of the sensitivity analysis can be found in Figure S6.”

See specific comments below:

13: I know you are focusing on long-term cooling effects, but perhaps you could at least mention (in the introduction) the potential short-term cooling effect of sulfur release from LIPs. See for example Landwehrs et al. (2020) or Black et al. (2018).

We have included these references and a sentence to detail this in the updated manuscript (L32-34):

“Greenhouse gas release may cause rapid warming⁷, with large-scale sulfate release driving short-term cooling⁸.”

19: Please specify what kind of model you used.

Change made specifying the SCION model (L15).

19: Could you please quantify this “minor effect” in your abstract? Include specific quantitative results from your study.

We quantify this in the updated version (up to 1°C of change).

25: I suggest this change: “The emplacement of large igneous provinces (LIPs1), systems of voluminous mafic magmatism related to processes other than seafloor spreading, has occurred regularly through Earth’s history2-4”.

Change made.

27: I would replace the word “enormous”. Also, please quantify the carbon release; e.g., up to xxx Gt; in the range of xx-xxx Gt C.

We have included an estimate of CO₂ degassing from the Siberian Traps here.

29: I suggest this change: “As a result, linkages between LIP emplacement and large-scale changes in the Earth system (e.g., environmental and climatic shifts) are often made^{3,7-8}, convincingly linking LIP emplacement to a number of mass extinctions^{2,3,9-11}”.

Change made.

43: I would use Davies et al. (2017) instead of Schaller et al. (2012) in this case. Marzoli et al. (1999) is a classical paper for linking CAMP and the ETE.

Change made, and Marzoli et al. (1999) also added here.

44: I would also cite Greber et al. (2020).

Reference added here.

46: I don't agree with this statement – the consensus is that carbon release is important, but cannot explain the mass extinctions alone. I would thus moderate this statement, and avoid saying that C release is a primary kill mechanism, because we actually don't know that.

We have tempered this sentence here (L49-50):

“It is generally assumed that one of the most damaging impacts of LIPs on the biosphere is carbon-rich volatile release...”

48: LIP emplacement into carbon-rich sediment is not really that rare; some other examples besides Siberian Traps are CAMP, Karoo, HALIP, Emeishan.

We have removed ‘rare’ here.

53: I would replace the word “holistic”.

‘Holistic’ replaced with ‘full’ here.

69: I would replace “make sense”.

Sentence adjusted ‘to test the validity of its predictions’.

70-71: Please rewrite this sentence.

Sentence rewritten: “By constraining the model against multiple proxy systems it is possible to make clear quantitative tests of hypotheses.”

74-77: Please rewrite this sentence.

Sentence broken into two and repetitive aspects removed.

80-81: How do Figs 2-3 clearly show that the inclusion of LIPs alters the model reconstruction of climate? What am I supposed to be looking at? Please be more specific.

We have adjusted this sentence to be more specific – this section focusses on the carbon cycling aspect of the model.

81-82: The input of carbon minus the removal from weathering - is this actually shown anywhere? Is this supposed to be what we see in Fig. 2b?

In the updated figure 2 this is now panel A, which we state in the updated manuscript.

81-84: This sentence seems out of place and like doesn't belong there. It's very broad. I would move to introduction or delete.

We have adjusted to make this less broad, and focus on the carbon-related aspect of the Earth system here.

85-87: I don't think Fig. 2 highlights the fact that weathering is the dominant driver. There is a lot going on in this figure. Am I supposed to be looking at 2b or c here? Please be more specific.

We have made it clearer in the updated manuscript that we mean here for the majority of the time period the carbon balance of LIPs is below zero, indicative of a net carbon drawdown of LIPs (L94-95):

"...with the majority of the studied time frame characterised by LIP-induced carbon drawdown, as indicated by carbon balance values below zero (Fig. 2a)"

Some other comments on Figure 2:

- I would make the legend font size bigger.
- What do you mean by "modern area"?
- What do you mean by relative LIP area versus present?
- What is the difference between the 0.05 Ma and 5 Ma degassing curves? This is somewhat explained further down in the discussion/methods, but you refer to Fig. 2 already in the beginning of the discussion, so the reader has no idea what these curves are supposed to represent. Either move them to the supplement, or include a description in the figure legend, or refer to specific text/table where you describe what these are.
- What does "Degassing Data" mean?
- I suggest to use different units of the y-axis for panel b and c. These values are difficult to read and they also make the figure look too busy, and it's hard to see where panel b/c starts/ends. Do you have to present your results in mol C?
- I don't completely understand what panel b is supposed to show. Is this supposed to be carbon input minus carbon removal? So, everything in the yellow area represents net removal and the opposite for the purple area? Can you perhaps think of another way to show this?
- Due to the range of y-axis in panel c, it looks like the carbon removal from weathering is constant (yellow line). This is an important part of your results. I think you should "zoom in" so we can see the variability.
- I suggest these following changes to Fig. 2 so that your most important results are highlighted: Move panel a to the supplement. Only show the degassing curve that you focus on in the main text and show the other curves in the supplement. Change the y-axis units of panel b and c. For the yellow curve, show a more appropriate range of the y-axis. For panel b, place y-axis ticks on for example the right side, and then for panel c, show the ticks on the opposite side. Or, could you instead combine the carbon balance in panel b and degassing

curve in panel c into the same plot? And then show the carbon removal in a separate panel instead.

These comments are very useful and we have updated figure 2 as suggested. Briefly, we have simplified the figure to present only carbon drawdown in panel C, and carbon release in panel B (with only two scenarios – high and low degassing). The updated panel A now displays the carbon balance value through time from the ‘high’ degassing scenario. All other information from this figure is now presented in the supplementary information.

92-93: Please refer to a figure to back up this statement.

Figure reference added here.

93-94: Where do I see that there is slight enhancement of net LIP-related CO₂ drawdown this in Figure 2?

Reference to figure 2c now added here.

93-94: I suggest this change: “Following the emplacement of the Siberian Traps (252 Ma), there is slight enhancement of net LIP-related CO₂ drawdown in the model (Fig. 2), reflecting the weathering of Siberian Traps material³⁶. However, the location of this LIP in the high latitudes (Fig. 1) suggests that the silicate weathering feedback is much weaker than if it were emplaced in the tropics³⁵”.

Sentences updated according to the suggestions.

97-98: If the reader does not read the methods section first, they don’t what you mean by “weathering-only scenario”. I think you should inform the reader about your different scenarios at the beginning of the “Result and Discussion” section. I suggest that you make a summary table that describes your different scenarios. Then you can just mention that you have x number of scenarios and refer to the table, and the reader can get more information there.

We have now added a table with all scenarios included in the supplementary information (Table S1). We have also added the following text to the manuscript (L87-91):

“To do this, we run a range of model scenarios (Table S1, Methods), these include a baseline (the unaltered SCION code), a model with only LIP weathering considered, and scenarios with only degassing, and with weathering and degassing considered. Unless otherwise stated, our discussion below refers only to the output of the scenario ‘Weathering & Degassing High’ (Table S1, Methods).”

99: Specify which panel in Fig. 4 that show these results.

Panel 4b highlighted here.

99-100: Which figure(s) show this? What do you mean by “original model”?

Figure reference added (Fig. 4d) and clarification that we mean the baseline model added here.

102-103: I can’t really see that CAMP leads to the highest LIP-related carbon burial values in Fig. 2.

We have adjusted Fig. 2 as suggested above, and so variability of this value is now much easier to see in Fig. 2c.

103-104: Figure 2 doesn't show the location of CAMP across the equatorial regions.

Adjustment made – we now only refer to Fig. 1 here.

110-112: What are you comparing to the baseline here? The weathering only scenario (blue line)? Please be specific. Also, please refer to the specific panel in Fig. 4 that you are describing (in this case 4c I think). It's really hard to see a 1°C shift because of the range of the y-axis in Fig. 4c; this is also the case for Fig. 3: I don't see a shift of 1°C in this figure. Please refer to Fig. 4d at the end of this sentence when you are referring to pCO₂.

We have included specific references to the model scenarios and figure panels we mean in the updated manuscript. As discussed below, we have also adjusted the axes for Fig. 4 so it is easier to see temperature variability.

114-115: "as seen in some proxy records in the early Jurassic³⁷": Here you refer to Dera et al. (2011), but Fig 4c show proxy data from Scotese et al. (2021). Why do you only show proxy records from Scotese in your figure? But then compare to Dera in the text? This question also goes for the rest of the panels of Fig. 4. They all have just one reference for the different proxy data. Why did you choose these proxy data as opposed to other records in other publications? I understand that you must make a decision, but it would be nice if you could discuss your choices.

For simplicity, and to ensure we have records which cover the whole period, we have decided to present only single proxy records in Fig. 4. The Scotese et al. (2021) record is selected as it covers the whole Mesozoic and provides an estimate of global temperature change. We feel it is suitable to compare to specific studies such as the one by Dera et al. (2011) in the text rather than clutter the figure with more data. We include a line in the Methods to explain our choice of proxy records (L381-386)

"To validate the model results, we compare to a range of geological and proxy data. We generally select the data based on two primary factors. Firstly, that the record covers the entirety of our modelled period, and secondly that it is global in scale. Using these criteria, we compare our model outputs to the global average temperature reconstruction of ref.⁴⁹, the global seawater ⁸⁷Sr/⁸⁶Sr curve of ref.⁷⁰ and compilations of atmospheric CO₂ (ref.^{71,72}) and carbonate δ¹³C (ref.⁷³)."

117-118: Again – which lines are you comparing? Which model results? Please be specific. Also, I am not sure if I agree with the general statement that "there is good correspondence between the seawater ⁸⁷Sr/⁸⁶Sr record and our model results". I see some correspondence for the green and blue lines between ca. 250 and 180, but that period represents only parts of your full study period.

We have clarified here the findings (L127-132):

"However, the good correspondence between the seawater ⁸⁷Sr/⁸⁶Sr record and our model results for the scenario 'weathering and degassing high' suggests the amount of additional unradiogenic material being input to the oceans is of the correct order of magnitude, at least for the period 250 – 180 Ma (Fig. 3)."

118-119: I don't understand how Fig. 3 shows that "the amount of additional unradiogenic

material being input into the oceans is of the correct order of magnitude”. Where in Fig. 3 am I supposed to see this?

We now point the reader to the location of these data: Fig 4b.

128-129: You state that the model is unable to reproduce most of the Mesozoic trends in oceanic Sr isotope composition prior to LIP addition and refer to Fig. 4. But don't you add LIP (Tarim/Skagerak) already at 300 Ma, so in the beginning of your study period? So, what do you mean by prior to LIP addition then? I would argue that, generally, your model is unable to reproduce Sr record prior to ca. 250 Ma and after 170 Ma.

We have adjusted this sentence here, to reflect our findings better. Instead of saying 'Prior to LIP addition', we now state 'Prior to the emplacement of the Siberian Traps...’.

130: delete “a”.

Change made.

136-140: Please rewrite this sentence.

We have rewritten (and broken up into 3) the sentence as follows:

“Further, Sr input resulting from enhanced continental weathering resulting from high global temperatures also improves the comparability of the data to the model outputs. This is particularly clear across the P–T boundary, where $87\text{Sr}/86\text{Sr}$ rises markedly. This rise is driven by the warming impact of the Siberian Traps and weathering of generally radiogenic crustal (non-LIP) material – as the LIP was emplaced at higher latitudes away from major weathering zones.”

146-148: Based on your results, how can you conclude/propose that volcanic ash is more important than basalt emplacement as a driver of CO₂ removal? What is the evidence behind this statement?

We have now updated this sentence. Rather than relying on our model results here, we are putting forward a separate mechanism to explain the link between cooling and LIPs, if weathering is unlikely to be a primary driver (the main finding of this work here) of global cooling, lines 163-172:

“One potential explanation is that the deposition of volcanic ash, known to drive periods of transient cooling¹⁵, may be more important than basalt emplacement as a driver of CO₂ removal³¹ on long time scales during LIP episodes. For example, during the late Ordovician, ash supply was sufficient to drive up to 3°C of cooling for up to a million years in a global biogeochemical model³¹. There was no LIP associated with the late Ordovician, but many LIPs had periods of intense explosive volcanism alongside the large-scale effusive eruptions. For example, the North Atlantic Igneous Province in the Cenozoic had an extended episode of basaltic ash deposition⁴⁶. This hypothesis requires further testing, but would explain how LIPs may still be linked with global cooling, despite the apparently small impact of their weathering on global climate.”

151-152: which model run uses this baseline degassing rate? If you make the summary table I suggested above, you could include that information there.

Clarified here as follows:

“The SCION model uses the baseline degassing rate of ref.³⁶ and in this work we add an additional component for contributions from LIPs in three scenarios (see Table S1, Methods).”

157-158: Please be specific when referring to the figures here; I assume you are referring to Fig. 4c, but which panel in Fig. 2?

We remove the reference to figure 2 here, and specify figure 4c.

158-159: Again, is this Fig. 4c?

Correction made.

159-161: How do I see that the scale of C release during ST emplacement is larger in Fig. 3?

Adjusted to make clear we mean Fig 2b (carbon release from LIPs).

160: Please replace the word “speed”.

‘Speed’ replaced with ‘rate’.

161-163: There is no red line in Fig. 3. Also, the baseline run should be a black dashed line right (not red)? What do you mean by the “new model scenario”? Please give all model scenarios clear names (list these in the table I suggested), and consistently use the scenario names throughout the text so that it is always clear which scenario/model run you are talking about.

Using the new scenario table we have clarified this here (and changed ‘red’ to ‘black dashed’ line for the baseline run, lines 185-188):

“In the ‘baseline’ SCION scenario, transient carbon cycle perturbations from LIPs are not included (Fig. 3, black dashed line), but in scenario ‘weathering and degassing high’, the P–Tr boundary is represented by a shift to pCO₂ as high as 10,000 ppm (Fig. 4d), reflective of proxy data from the period¹⁷.”

163-164: I think you are referring to Fig. 4d here? Also, here you refer to proxy data from Wu et al. (2021), but panel d in Fig. 4 includes proxy data from Saltzman & Thomas (2012). Again, why do you compare to proxy data that is not in the figure, and why is this proxy data not in the figure? Why did you choose to present only Saltzman & Thomas (2012) in your figure?

As discussed above, for figure simplicity and clarity, we chose to present only long-term global synthesis data in Figure 3. We think it is sometimes useful to discuss our findings in the light of other work, even if we do have space to present that other work in our figure.

164-165: Are you referring to your own model results here or to another study? Please specify.

Specified here that it is our modelling results.

170: Again, it is very hard to see a 4°C rise in Fig 4c due to the range of the y-axis.

As suggested, we have altered the y axes on this figure.

174-175: Which results? Please be specific. Also, you refer to “previous proposals”, but mention only one reference.

We now specify we mean our results relating to temperature change. We have also now changed this sentence, and removed ‘previous proposals’ to make it clearer.

177-179: Please add a more thorough description of this additional run in the methods section. Again, if you make a summary table, you can include information there. You say “reduce the degassing window to 50 ka for each even”; what is the original “degassing window”? And where is this described? I can barely see the dashed cyan line in Fig. 4c (again, please refer to the correct panel). I would suggest to change the color, and/or “zoom in” and change the range of the y-axis. Also, please be consistent, use either 50 ka or 0.05 Ma (as shown in Fig. 2 or in the methods; line 302).

We have removed the results of this modelling from Figure 4 and moved them to the supplementary information.

Some other comments on Figure 4:

- Add grey dots and shading to the legend and name it “proxy data”.
- I don’t think panel a is discussed at all in the main text? So, you could move this to the supplement (or discuss it, if you think these results are important).
- It’s really hard to see the different model results, and compare the different runs, because the range of the proxy data is so much larger than your results (y-axis). I understand that you want to show the range of the proxies, but your model results are the most important. I suggest that you “zoom in” and focus on a y-axis range that is more appropriate for showing your results. This would cut off the proxy data, but you could show the full range in the supplement.
- I recommend to change the cyan color because it is very hard to see, although this problem may be solved if you change the range of the y-axis.
- From looking at this figure, the difference between the two “Weathering & Degassing” scenarios is not clear. If you make a summary table, you could refer to that in the figure caption.
- In the figure caption you say upper panel, below, third and final panel, but why not refer to a,b,c,d?
- Please thoroughly describe in the discussion: 1) why you chose the specific proxy data shown in this figure, and 2) why your model curves don’t match the proxy data in all four panels.
- Are the green and cyan lines here equivalent to the red dashed and red dotted lines in Figure 2? Same question for the blue (weathering only) line in Fig. 4 and the yellow line (weathering) in Figure 2. If so, please be consistent with both the color/type of line and the name of the model run.

We have taken on board all comments listed above in the updated figure 4, and updated the figure caption to reflect the changes. We appreciate the suggestions and think it’s a much clearer figure now.

180: What do you mean by “previous reconstructions”?

Adjustment made (L208-214):

“In this scenario (‘Weathering & Degassing 50 kyr’), the temperature rise across the P–T is 9°C, and for the CAMP it is 8°C, much closer to the reconstructions of 15°C for the P–T^{48,49}

and >5°C for the CAMP emplacement (ref. ⁵⁰ Fig. S4),), suggesting carbon volume estimates may be correct, but the periods of degassing may have been much shorter than the lifetime of the LIP.”

183: I would replace the word “purport”.

‘Purport’ replaced by ‘claim’.

190: Another important limitation is the uncertainty from original LIP volume/extent and magnitude of carbon degassing from the LIPs, but this is not discussed. Please add that to this part of the discussion.

We now include a discussion of this in the relevant section limitations (L263-270):

“For estimating rates of carbon degassing from LIPs, it is vital for accurate reconstructions of original LIP volume and rate, as well as any cryptic degassing component of LIP emplacement⁵⁴. Considerable work has been completed to understand these features this, but due to subsequent erosion and subduction, volume estimates will always be associated with high errors (see Methods and Table S2 for the range of degassing estimates used here). Approaches to reconstruct emplacement rates are improving, and with ongoing methodological improvements in radiometric dating, we anticipate future work to further improve and constrain these estimates.”

204-222: These sections seem out of place. You kind of jump back and forth between limitations and implications (the first two and last paragraphs discuss limitations). Perhaps re-structure this section into “implications” and “limitations” separately.

We agree, and have split this section into two, moving all paragraphs to the correct locations.

231: Please add a reference.

Reference added (Mills et al., 2021).

238: What do you mean by “whole ocean”?

“Whole” changed to “Global” here.

248: I suggest this change: “Ref.30 calculated a LIP half-life of 29 Ma (Equations 1 and 2), allowing us to calculate a fractional area of each LIP at a time after its emplacement (Adecay):”. Then after “... the calculated half-life (=29 Ma).” add “Note that the half-life increases to 36 Ma if one accounts for burial of LIPs³⁰, but this is not taken into account here”.

Adjustment made.

260-262: Just to clarify: does this mean that 1 = the maximum extent of the LIP, and 0 = 100% of the LIP has eroded?

Yes – we clarify this in lines 303-304 now:

“...and update the relevant cell to reflect this (so after emplacement LIP cells will be between 0 and 1, with 1 representing 100% LIP extent).”

284-287: Instead of using Siberian Traps as an example, list all the LIPs where this is true

and you used a value from previous literature. And again, explain why you chose that specific degassing value (as opposed to other estimates, or for example the mean of a range).

As discussed in response to the major comment above, we have adjusted this section of the methodology in lines 334-355:

“For modelling purposes, these values are converted to molar carbon flux. For each LIP, we derive two degassing scenarios, ‘low’ (calculated using the lowest volume and longest duration in the literature) and ‘high’ (calculated from the highest volume and shortest duration in the literature). Some LIPs are emplaced into carbon-rich rocks, a process which leads to additional carbon degassing through thermogenic alteration. Where estimates of this process are available, for the CAMP⁶⁵, Karoo⁶⁶ and Siberian Traps⁶⁷, we use those values for our ‘high’ scenario (see Table S1 for further information).

For each LIP, the rate estimate is used to emit CO₂ into the model atmosphere. We use Gaussian curves to complete this, with the midpoint of the LIP activity and the peak of carbon emissions used to construct the function (see Table S2). We set the width of the Gaussian function to be related to the period of activity known for the LIP. For the ‘low’ scenario, we construct the curve based on the longest possible duration, and for ‘high’ the shortest. For example, the Siberian Traps is taken to have degassed in 2 Myr⁶⁸ in the ‘low’ scenario and in 0.8 Myrs⁶⁹ in the ‘high’ scenario, see Supplementary Table 1 for all LIP details. In addition to the ‘low’ and ‘high’ scenarios, we run a further configuration with the full impact of cryptic degassing of the Siberian Traps included, using the approach of ref.⁵⁴. This configuration (‘high & cryptic’) considers a more complex approach to estimating the total degassing of a LIP including degassing which continues after the emplacement of lava has finished⁵⁴. For carbon isotopic mass balance, we include a new ‘LIP CO₂ δ¹³C’ estimate in the model, which we take to be -5 ‰, and is the isotopic composition of the degassed CO₂. All modelled configurations are detailed in Table S1, and the ranges of input data relating to these configurations are in Table S2.”

287: Also list the LIPs where you calculated this yourself. I know that this information is provided in Supplementary Table 1, but you can easily just list the different LIPs here, and then refer to the table for more information.

We refer the reviewer to our response to the previous comment here, and include all necessary data in the updated supplementary table (now Table S2).

298-299: Does all literature really agree that the Siberian Traps and Karoo degassed in 1 and 0.47 Myrs, respectively? There are more recent papers than those cited, for example Burgess et al., (2017), Kasbohm et al. (2021), Greber et al. (2020), Gaynor et al. (2022). The same question goes for the other LIPs. The rate of degassing is important when constraining climatic and environmental effects. Please explain why you chose these specific references/values.

Again, we refer the reviewer to our initial response to their comments earlier in this document. We have used the literature cited and a more detailed search to update all the estimates of degassing.

300-302: What do you base the duration of 0.05 Ma on? Why do you refer to Fig. 1 here? Give this model run a specific name, and be consistent throughout the manuscript and in the figures.

The duration is an arbitrary value used to represent a short degassing period (and therefore rapid degassing rate) for each LIP. We include a new description of this scenario here (L368-372):

“An additional test runs a model where all degassing episodes are assumed to have only 50 kyr duration (an arbitrarily short time to drive rapid carbon degassing rates), to investigate the ability of rapid carbon release to drive climatic warming (scenario “Weathering & Degassing 50kyr”). All model scenarios are outlined in Table S1.”

315-318: please describe this sensitivity test more thoroughly. What does “random variation of a number of variables following previous work” mean? Which variables? Are the results from this sensitivity test only shown in Sup. Fig 2? What do the results tell us? Sup.Fig.2 is lacking a legend, so it's hard to understand what I'm looking at here.

We now include a better description of the sensitivity analysis here (L375-380):

“Specifically, the weathering enhancement from plant evolution is varied between no enhancement and 7-fold enhancement, the carbon isotope fractionation associated with microbial sulfate reduction between 20–40‰ and the fractionation associated with photosynthesis between 20–30‰ (land) and 25–35‰ (marine). See ref. ³⁶ for further detail on SCION sensitivity tests. Results of the sensitivity analysis can be found in Figure S6.”

332: Mean of what? The 1000 simulations? Does the green line shown in Figure 4 represent a mean of the 1000 simulations?

Figure 4 displays single simulations not ensemble runs. Here we clarify this point, and point the reader to where these results are found – Figure S4.

Figure 3: This figure is barely mentioned. Maybe move to supplement? Also, add a,b,c to the different panels.

As we believe this is a nice demonstration of the spatial variability of climate we retain its use in the text. We now refer to it more in the discussion.

Fig. S1: Add legend. I would change the yellow color – it's quite difficult to see.

Legend added here. Colours of lines have also been adjusted.

Supplementary Text: “This value is then multiplied by the LIP multiplies (in this case 7)”. Note it should be “is” and not “in”. Also, could you please remind the reader why 7?

Change made, and addition here to clarify why we use a value of sevenfold – it is based on previous estimates (Dessert et al., 2003) and previous work (Longman et al., 2022, Lefebvre et al., 2013).

Reviewer #2 (Remarks to the Author):

The impact of Large Igneous Provinces (LIPs) on climate is widely acknowledged in scientific research, yet it remains largely untested. With the potential to cause both warming and cooling effects, LIPs serve as an ideal mechanism to explain puzzling climate shifts. At least two reasons make this study valuable and interesting:

1. It specifically tests the effect of LIPs on the long-term carbon cycle and climate change

during a period with multiple LIP episodes.

1. It demonstrates a limited but significant effect, with results validated by strontium (Sr) simulations that align with geological records, providing constraints on its reliability.

This manuscript tackles an essential and engaging topic, numerically assessing the impact of LIPs on both the degassing and weathering components of the carbon cycle. The study's scientific importance is significant, especially given the widely hypothesized but still underexplored role of LIPs in climate shifts and mass extinctions. Furthermore, it addresses the current knowledge gaps regarding the true climatic potential of LIP events.

In terms of scientific quality, the study is rigorous and well-executed. The authors thoroughly test the reliability of their methods and provide detailed discussions of their approach. While additional tests, such as reverse modeling and climate sensitivity analyses, could further solidify the conclusions, the current methodology is robust and well-supported. The scientific reproducibility of the study is strong, with the methods clearly explained and the data presented in a way that allows for replication.

The data and methodology used in this study are solid, with justifications for the time interval and LIP selection, models employed, and acknowledged limitations. This thorough approach enhances the credibility of the results, making the study a reliable source for further exploration of LIPs' climate impact.

The analytical approach is equally strong, providing a clear framework for understanding the relationship between LIPs and the long-term carbon cycle. However, some clarity is needed in the methods section, particularly regarding the link between the General Circulation Model (GCM) simulations and the LIP effects, which remains somewhat unclear. Additionally, the effect of the Siberian Traps could benefit from a more nuanced treatment, especially in light of evidence suggesting that its influence may not strictly align with its location outside the tropical zone, indicating a potentially more complex, non-latitudinal distribution of its climatic effects.

The validity of this study is well-established, with its methods, findings, and conclusions all supported by thorough analysis. The presentation is effective, with figures that enhance the text and aid comprehension.

Addressing the minor points mentioned—particularly in improving the clarity and comprehensiveness of the manuscript—will strengthen the overall contribution. In conclusion, I recommend this manuscript for publication with only minor revisions.

We appreciate the reviewer's support for our work and their helpful comments. We have attempted to address all concerns in the updated manuscript.

Specific comments: (the numbering applies to the version given to the reviewer file)

(abstract) It would be beneficial to highlight that the implementation of LIPs in the model results in a better alignment between simulated Sr levels and the geological record, suggesting that this approach is plausible and reliable.

We have included the following in the updated abstract (L17-19):

“This approach results in better correspondence between the modelled output and proxy reconstructions of the period (especially for seawater Sr isotope composition).”

(65-67) You mention the prevalence of low-latitude aridity in Pangea, yet none of the maps in Figure 3 show significant arid regions developing in the tropics. Is this discrepancy intentional?

Yes, it is more correct to say subtropical aridity. We have revised the paper to simply state that aridity is extensive, as that is the main point of the section.

(72-74) It remains unclear to me, even after reviewing the entire study, whether changes in topography related to the implementation of the LIPs were incorporated. If so, what is the magnitude of these changes? Additionally, were any GCM simulations rerun for the time slices where LIPs were included? As we know, all the volatiles associated with LIPs could also influence the hydrological cycle.

We have made it clearer in the revision that we do not adjust the topography in this study in response to LIP emplacement. We update the manuscript to clarify this (L74-78):

“As well as incorporating volumes of degassed CO₂ during emplacement, these LIPs are added to the 2D model land surface grid as basaltic terranes and interact with local temperature, relief and hydrology to amplify silicate weathering rates (Fig. 1), although the LIPs themselves have no topographic height so the model topography is not altered by LIP emplacement.”

GCM simulations were not rerun for timeslices associated with LIP emplacement, but this will be tackled in future work. Indeed, the role of volatiles is not properly addressed in this study, beyond the simple degassing of carbon into the atmosphere box.

One simple way of considering the impact of LIPs on the physical topography (i.e. increased height due to basaltic emplacement) is to enhance the weatherability of LIP material, as a proxy for this process. In the SI we now include the results of a 30x enhancement run, which demonstrates that at this level the Sr isotope record is invalidated. We also discuss the implications of this in the updated manuscript.

(81-84) Regarding Fig. 2, how do you define tropical land? Are specific latitudinal boundaries used? It's important to note that during the Pangea period, latitude-based estimates of tropical land may not be accurate due to the continental configuration of central Pangea. Could you elaborate on how this was addressed in your analysis?

In light of the comments of reviewer 1, we have now moved the panel relating to tropical land to the supplementary information (Fig. S1). We do now include a definition (up to 20° from the equator) in the new figure caption. We also highlight this estimate is purely based upon the total number of gridsquares which fall in this region and contain LIPs.

(93-96) This assertion holds true only if we assume that climatic belts were similar to those of the present, which may not always have been the case. For instance, examining between 270 & 250 Ma the distribution of bauxites and laterites, which are indicators of 'tropical conditions,' reveals a shift towards higher latitudes, as well as a tendency for these deposits to be located on isolated terranes. On the other hand, evaporites tend to cover most of the tropical region. Could you discuss how this perspective was considered in your analysis? (similar to previous comment I guess).

As discussed above, we use a very simple definition of what is 'tropical' in this study. The reviewer is quite correct to point out that this is not fully suitable for the period studied, but for simplicity, and for continuity through our modelled period we retain the definition of <20° from the equator.

(115-117) This presents quite a challenge, as you are attempting to define this effect. Could you try to constrain this underestimation? Specifically, how much would you need to increase the weatherability to achieve the 5°C cooling? Some reverse modeling could help clarify this.

This is a great suggestion, and we have completed some extra modelling runs with enhanced weatherability to test it in a simple manner (reverse modelling is a good suggestion for follow up work). To reach the full 5°C of cooling, we require 30-fold weatherability enhancement. We include a discussion of this point in the updated manuscript (L130-136):

“Indeed, to reach 5°C of cooling following the CAMP emplacement, the weathering enhancement value must be raised to thirty-fold (Fig. S2b), a value which invalidates the Sr isotope curve, resulting in values well below the reconstructed proxy data (Fig. S3b). A weathering enhancement of fifteen-fold remains possible given the Sr isotope curve across the CAMP (Fig.S3), suggesting it may be a plausible maximum value for this LIP (and one which results in 2°C of cooling), but this would require considerable uplift of basaltic material”

(117-119 + 129-131) This is great and needs to be also said in the abstract.

As suggested above, we now mention the correspondence of the Sr curve in the updated abstract.

(140) Higher altitudes ? I guess you meant latitudes?

Change made.

(140-148) While I agree with your conclusions, I think there might be more to elaborate on in this paragraph. I noticed that the 220 Ma and 200 Ma maps included in SCION show little to no topography compared to the other maps. Could this lack of topographical variation pose a problem for the simulations and potentially impact the effects of LIP weathering? You also suggest volcanic ash as a cooling driver. Is it possible to test this in the model as well?

This is a good point. We have noted that the more muted topography in these maps may be a factor in the limited weathering of LIP basalts in the limitations section (L259-261):

“For example, the muted topographical variation of our plate reconstructions between 220 – 200 Ma may be a factor in the limited weathering of LIP material at this time (Fig. 3c).”

The question of ash impacting climates is one we intend to tackle fully in upcoming work. A single example of the approach can be seen in Longman et al. (2021), and we note this in the revised paper, and include a little more detail (L162-169):

“One potential explanation is that the deposition of volcanic ash, known to drive periods of transient cooling¹⁵, may be more important than basalt emplacement as a driver of CO₂ removal³¹ on long time scales during LIP episodes. For example, during the late Ordovician, ash supply was sufficient to drive up to 3°C of cooling for up to a million years in a global

*biogeochemical model*³¹. There was no LIP associated with the late Ordovician, but many LIPs had periods of intense explosive volcanism alongside the large-scale effusive eruptions. For example, the North Atlantic Igneous Province in the Cenozoic had an extended episode of basaltic ash deposition⁴⁶.”

(167-168) Why not conduct additional simulations with varying climate sensitivity to determine if this is indeed the cause of the disconnect?

This is not something we can test easily. The climate sensitivity referred to here is that of the FOAM model used for the GCM runs. As such, to test variations in climate sensitivity, we would need to perform a large number of runs using an alternative high-sensitivity GCM, which is a difficult project in itself. We do now clarify that this sensitivity issue relates to the GCM, not to SCION per se in the updated manuscript.

(174-180) Similar to the approach with climate sensitivity, it would be valuable to assess whether the degassing required to induce such a temperature rise is plausible. This is somewhat addressed by reducing the degassing window, as it effectively increases the rate of degassing per time unit.

As discussed in the response to reviewer 1, we have updated the manuscript to clarify that in the models with shorter degassing windows temperature rises in line with proxy data can be reconstructed for the P-T and the CAMP. Further, we now also present the results of a model which considers in detail the role of cryptic degassing (from Black et al., 2024), which allows us to reconstruct the full warming of the Siberian Traps. We have added this information to the updated manuscript (L205-214) and in two new figures (Fig. S2, S3):

“By running comparative models which reduce the length of time the degassing occurred for to 50 kyr for each event, we can reconstruct changes in climate which are much closer to proxy reconstructions (Fig. S4). In this scenario (‘Weathering & Degassing 50 kyr’), the temperature rise across the P–T is 9°C, and for the CAMP it is 8°C, much closer to the reconstructions of 15°C for the P–T^{48,49} and >5°C for the CAMP emplacement (ref.⁵⁰ Fig. S4), suggesting carbon volume estimates may be correct, but the periods of degassing may have been much shorter than the lifetime of the LIP. A further model scenario which fully considers the role of cryptic degassing (Weathering & Degassing Cryptic), using the approach of ref.⁵⁴, is also able to reconstruct up to 15°C warming for the P–T (Fig. S2a).”

(197-199) While you emphasize this point here, it's unfortunate that nothing regarding this aspect is mentioned in the abstract, as it just makes the study's reliability stronger.

As mentioned above, this is now included in the updated abstract.

(207-212) However, the tests conducted here do not account for changes in climate sensitivity, and some of the maps exhibit very little or no topography. This point may need to be addressed with more nuance.

We include a qualifying statement at the end of this paragraph in the updated manuscript (233-235):

“However, further tests considering in more detail important aspects of the SCION model such as climate sensitivity must be completed to be certain of these conclusions.”

We also include a sentence on topography as a limitation (L259-261):

“For example, the muted topographical variation of our plate reconstructions between 220 – 200 Ma may be a factor in the limited weathering of LIP material at this time (Fig. 3c).”

(218-219) Very true ! it is a very important statement to make.

Thank you for agreeing!

(223-227) Yet, It is very unlikely this belt remained located $\pm 10^\circ$ NS latitudes (see 270-250 Ma evaporites distribution).

We agree with this point, and that is why this section is now included in a specific ‘limitations’ section of the manuscript. Further work should be focussed on improving questions relating to this sort of question.

(262-265 +272) Yes, manipulation is required; however, how are temperature and runoff modeled in relation to the LIPs? This aspect is not very clear in the methods section.

As discussed above, the topography of the LIPs is not considered in the GCM runs, and we have clarified this in the updated manuscript.

Full details of the equations governing the interaction between runoff, temperature and erosion may be found in Mills et al. (2021), and we do not feel the need to repeat this information here. Instead, we now point the reader here to that publication for further information (L322-324):

“ The equations governing the relationship between temperatures, runoff and erosion are those used in ref.³⁶, and we refer the reader to that publication for more detail”

(Figure 1) This is a very nice figure, but it feels a bit crowded with names. Perhaps consider using just the first letter in the figure itself and providing the full names in the figure description.

Thank you for the suggestion – we have made this change.

(Figure 3) I have questions primarily regarding the 260 Ma maps. When surface air temperatures reach 30+ degrees Celsius across nearly $\pm 30^\circ$ N/S, you observe very high runoff and weathering. How is this possible?

Such high temperatures at high latitudes would indeed likely impact runoff (with evaporation being concurrently high). If these temperatures were uniform across the high latitudes we agree runoff would likely be low, but temperature across these regions is not uniformly high, with variability, and so the possibility for high runoff.

(Figure S1) What do the yellow and orange colors represent? I assume they indicate the mean and the upper/lower limits from the sensitivity test?

As discussed in the response to reviewer 1, we have clarified this in a new figure caption.

(Figure S2) This is a fantastic figure! It significantly helps in understanding the weathering implementation.

We appreciate the comments.

I sincerely appreciated the opportunity to read this study, which I believe makes a significant contribution. Thank you for allowing me to review it.

Reviewer #1 (Remarks to the Author):

Please find our responses to the comments below, highlighted in blue. We attach a version of the manuscript with changes tracked. For our responses below, however, we use line numbers pertaining to the updated (changes accepted) manuscript.

Second review; Longman et al; Limited long-term cooling effects of Pangaeian flood basalt weathering.

The introduction gives a clear overview of background information, what the authors have done and why. The discussion and description of results is now much easier to follow as the authors have improved both the text and figures. The additional model runs, analyses, table and discussion make this study much more robust and convincing. I would recommend publication. See some minor comments below (line numbers are from the document including the tracked changes):

We appreciate the reviewer's comments and have attempted to address all their final concerns in the updated manuscript. Our responses are detailed below.

18: Could you please specify "better" correspondence compared to what?

We have updated this on lines 18-20:

"This approach results in better correspondence between the modelled output and proxy reconstructions of the period (especially for seawater Sr isotope composition) when compared to previous modelling studies."

80-81: I suggest moving the following sentence to methods: "although the LIPs themselves have no topographic height so the model topography is not altered by LIP emplacement"

We have moved this sentence to the methods (lines 327-328).

123: Please check if the reference to Fig. 3b and c is correct for this sentence.

This has been adjusted to reflect silicate weathering is in panel 3c. We also add panel letters to figure 3.

127: Specify the panel letter of Figs. 3,4

Panels (Figs. 3a and 4c) now specified here.

136: The addition of Fig. S2 is great! But I recommend some changes:

- Adjust the y-axis of panel b of figure S2 ("zoom in") so that we can more easily see the temperature changes from the baseline. I get it that you want to show the full proxy range, but we can see that in panel a.
- Combine panel a of S2 and Fig S4 into one figure. That way we can easily compare all model runs. Get rid of all the white space; we want the focus to be on your model results.
- Place the black dashed line on top so we can see the full extent of the baseline run (it's kind of hidden behind the solid lines as of now). This applies for all these figures + Figure 4 in the main manuscript.
- Combine the a and b panels of figure S3.

We have made all suggested changes, thank you for the help refining the figures.

150: This is from my previous review and your response:

128-129: You state that the model is unable to reproduce most of the Mesozoic trends in oceanic Sr isotope composition prior to LIP addition and refer to Fig. 4. But don't you add LIP (Tarim/Skagerak) already at 300 Ma, so in the beginning of your study period? So, what do you mean by prior to LIP addition then? I would argue that, generally, your model is unable to reproduce Sr record prior to ca. 250 Ma and after 170 Ma.

We have adjusted this sentence here, to reflect our findings better. Instead of saying 'Prior to LIP addition', we now state 'Prior to the emplacement of the Siberian Traps...'.

I agree with your response, but I don't see this change in the revised manuscript (line 150).

Thanks for noticing – we have made sure this addition is in the updated manuscript.

156: Please check if the reference to Fig. 4c is correct for this sentence.

Figure reference updated, no Fig. 4b.

196: Please check if the reference to Fig. 3 is correct for this sentence.

Figure reference updated, now Fig. 4.

220: It's impossible to compare the 'weathering^oassing high" with the "50 kyr" run because the red line is hidden. Could you please make one of the lines dashed. Also, (I already mentioned this above), but please adjust the y-axis and/or get rid of the white space so it's easier to see the change in temperature.

As mentioned above, these have all been combined into two panels in Fig. S2, with the suggestions included.

231-232: This sentence seems strange (i.e., while when and clear driven).

Changes made.

247-248: This reads a bit strange, so consider re-writing: "...further tests considering in more detail important aspects..."

Change made, 'in more detail removed' here.

271: I don't really understand what you mean by "falsified by the geological record", so consider re-writing this.

Changed to 'would not be in agreement with the geological record'.

284: Delete "this" after "features"

Change made.

Fig 3: Could you mark CAMP and the Siberian Traps with a symbol instead of the dotted line? You could make the symbol transparent so we can still see the colorbar. I wonder if you should also add letters a,b,c (and update the text accordingly).

We do not feel that adding a symbol to these figures would make them clearer, and so retain the current line used to indicate their approximate location. We have, however, added panel letters and updated text to indicate which panel is being referred to.

Line 410: This is from my previous review and your response:

Fig. S1: Add legend. I would change the yellow color – it's quite difficult to see.
Legend added here. Colours of lines have also been adjusted.

I don't see these change in the revised manuscript (Figure S1 is now S6).

Correct figure uploaded in updated submission.